# Lead halide perovskites for photocatalytic organic synthesis

Xiaolin Zhu [1], Yixiong Lin[1], Jovan San Martin[1], Yue Sun[1], Dian Zhu[1] & Yong Yan [1]

Nature is capable of storing solar energy in chemical bonds via photosynthesis through a series of C–C, C–O and C–N bond-forming reactions starting from $CO_2$ and light. Direct capture of solar energy for organic synthesis is a promising approach. Lead (Pb)-halide perovskite solar cells reach 24.2% power conversion efficiency, rendering perovskite a unique type material for solar energy capture. We argue that photophysical properties of perovskites already proved for photovoltaics, also should be of interest in photoredox organic synthesis. Because the key aspects of these two applications are both relying on charge separation and transfer. Here we demonstrated that perovskites nanocrystals are exceptional candidates as photocatalysts for fundamental organic reactions, for example C–C, C–N and C–O bond-formations. Stability of $CsPbBr_3$ in organic solvents and ease-of-tuning their bandedges garner perovskite a wider scope of organic substrate activations. Our low-cost, easy-to-process, highly-efficient, air-tolerant and bandedge-tunable perovskites may bring new breakthrough in organic chemistry.

[1] Department of Chemistry and Biochemistry, San Diego State University, San Diego, CA 92182, USA. Correspondence and requests for materials should be addressed to Y.Y. (email: yong.yan@sdsu.edu)

The intentional construction of organic compounds via cost-effective and efficient photocatalysis is highly desirable. Remarkable advances in artificial C–C, C–O, and C–N bond formations have been made, including the development of protocols to merge photoredox catalysis with organic[1], transition-metal catalysis[2], and inorganic semiconductors[3–5]. However, many current catalysts require: high-cost noble metals; complicated synthetic preparations; air-free reaction conditions; or demonstrate moderate activity and are thus, not desirable[1–6]. A need to develop easy-to-produce, economical, effective and highly-tolerant photocatalyst for a broad scope of chemical bond formations, remains a significant challenge. Of the potential photoactive materials, Pb-halide perovskites APbBr$_3$ are an attractive candidate[7]. They have shown promise for low cost solar energy conversion (e.g., they have strong light absorption[8], long excited state lifetimes[8], efficient separation and transport of opposite charge carriers[9,10]). As a result, revolutionary advances have been claimed in perovskite photovoltaics, i.e. PCE has reached greater than 24.2% in only a few years of development[7,11,12].

Given the widespread success of perovskite in both efficient charge separation and electron-hole diffusion (length > 175 μm)[9], we recently questioned whether it might be possible to apply this unique material that has been proved in photovoltaics towards highly efficient photocatalytic organic synthesis. In photovoltaics, the absorption of photons induces the creation of electron/holes, while in photocatalysis, the equivalent is the production of reducing/oxidizing charges that can drive the desired chemistry. For photocatalysis, such reducing/oxidizing equivalents (excited electrons/holes) should live long enough and be transported efficiently to a catalytic site where chemistry occurs (i.e., at the photocatalyst surface). Therefore, photophysical properties of Pb-halide perovskites demonstrated for photovoltaic applications, also should be of interest in photocatalytic organic synthesis[13–17]. We recently demonstrated that the intrinsic surfaces of MAPbI$_3$ and MAPbBr$_3$ perovskites have low surface recombination velocities[8,18–20] indicative of an intrinsic low surface defect density that would otherwise hinder surface chemical reactions needed for photocatalytic systems. Our initial exploration of perovskite towards photocatalytic α-alkylation of aldehydes successfully proved that C–C bond formation reactions are efficiently achievable[17]. Other organic reactions focusing on styrene polymerization[15], benzenethiol dimerization and C–P bond formation between tertiary amines and phosphite esters[16] were also reported. We also note that a perovskite-based photocatalyst cell, perovskite/TiO$_2$ or NiO$_x$/perovskite/TiO$_2$ is report to photo-oxidize benzylic alcohol or activate C(sp$^3$)-H bond, although the yield is low, ranged from 0.016% to 0.73%[21,22]. Presently, it is still unknown if perovskites can make a general impact on organic synthesis.

Here we show that C–C, C–O, and C-N bond formations that are of fundamental significance in drug development and materials synthesis, are realized via perovskite nanocrystals (NCs) in high yield under visible light. Perovskites' unique role towards charge separation and transfer in photocatalytic reactions has been illustrated. Key concerns on perovskite as a photocatalyst, i.e. size, stability, reaction condition tolerance, and key catalytic metrics have been discussed. Moreover, band-tuning of perovskite using halide-exchange has been experimentally employed to activate previously unachievable reactions.

## Results

### General acceptance of perovskites for organic synthesis.
Perovskite colloidal suspension (CsPbX$_3$: $E_{CB} = -1.2 \sim -1.4$ V, $E_{VB} = +0.6 \sim +1.5$ V, all vs SCE; CB: conduction band; VB: valence band)[23] are effective catalysts for several fundamental organic reactions under visible light as shown in Fig. 1. Direct C–C bond formations are observed via C–H activation of aldehydes (**1a**, **1b**) or tertiary amines (**1c**, **1d**). The scope of the former reaction is not only limited on previously explored C–Br weaker bonds[17], but also covers stronger C–Cl bond. The absence or presence of oxygen is the key to lead to chain-extension product (**1c**) or an unexpected cyclization reaction (**1d**). C–N bond formations via direct N-heterocyclizations forming pyrazoles (**2a–f**) and pyrroles (**2g–i**), critical reaction for pharmaceutical development, are realized in high yield with perovskite at room temperature. C–O bond formation via aryl-esterification (**3a–f**) was achieved with a Ni co-catalyst. The respective reaction conditions are also optimized with regards to solvents, types of perovskites, air-tolerance, co-catalysts, and reaction time, etc. (see Supplementary Tables 1–8 for details). Catalyst loading has also been explored (Supplementary Tables 1–7) and respective minimum loading for typical reactions of ~0.1–0.5 mmol has been listed in Fig. 1. These reactions result in respective products in moderate to high yields without need for anaerobic sparging. The scopes of each aforementioned reaction were explored with various functional groups. (Fig. 1 and "Methods" section for details) As expected, control experiments reveal no product in the absence of photocatalyst or light.

**Perovskite's size effect**. The perovskite colloids, **P1**, described above are readily synthesized according to previous report[17,24] via directly mixing of readily available low-cost starting materials, PbX$_2$ with CsX, in an open vial under bench-top conditions (Supplementary Fig. 1). The resulting gram-scale emissive perovskite colloids exhibit a broad size-distribution, ca. 2~100 nm (Fig. 2a). The observation together indicates a bandgap energy of 2.4 eV that well matches the bulk CsPbBr$_3$ bandgap[7,25]. The synthesized colloids are too large to be in the quantum-confinement regime (Fig. 2a). Thus, for the system we are considering most colloids within the ensemble are larger than the Bohr radius, and hence the bandedges are determined by bulk bandedges and quantum-confinement effects do not contribute.

In contrast, using a high temperature synthetic method[26,27], we also synthesized size-controlled CsPbBr$_3$ NCs (**P2** 14 nm, $\lambda_{PL} = 521$ nm; **P3**, 9 nm, $\lambda_{PL} = 515$ nm; **P4**, 6 nm, $\lambda_{PL} = 508$ nm; **P5**, 4 nm, $\lambda_{PL} = 467$ nm, Fig. 2b–d and Supplementary Fig. 2). As shown in Fig. 2d, these NCs show a blue-shift probably due to quantum confinements[26,27]. The photocatalytic ability has also been explored in the same reaction condition. In C–H activation, at the early stage of the reaction, we find that smaller size NCs, i.e. **P2–P4** show a higher initial reaction rate compared to the original synthesized **P1** NCs. (Supplementary Fig. 3). However, small size NCs' catalytic reactivity diminished quickly. When breaking a C–Br bond to form **1a**, the reaction yield is recorded as 54–64% using **P2–P4** in less than 40 min, and longer reaction time leads to a marginal increase of the yield of **1a**. Much lower yield, ~8% was observed within **P5** probably due to a significant blue-shift leading to less visible absorption. Whereas using **P1**, the reaction rate is slower, however, the yield continuously increases and reaches 85% in ca. 5 h.

We suspect that small size NCs have higher surface area-to-volume ratio (Supplementary Table 9), hence a faster rate at the early stage. However, detrimental effects, i.e. moisture residue in solvent are inevitable. Such effects are more prominent on small size NCs than **P1**. We assume if the desired photocatalysis is slower than perovskite decomposition, the reaction yield may be of significant discrepancy between small and large size NCs. Such assumption is corroborated with reaction **1a** described above. In contrast, if the decomposition is not prominent, the yield

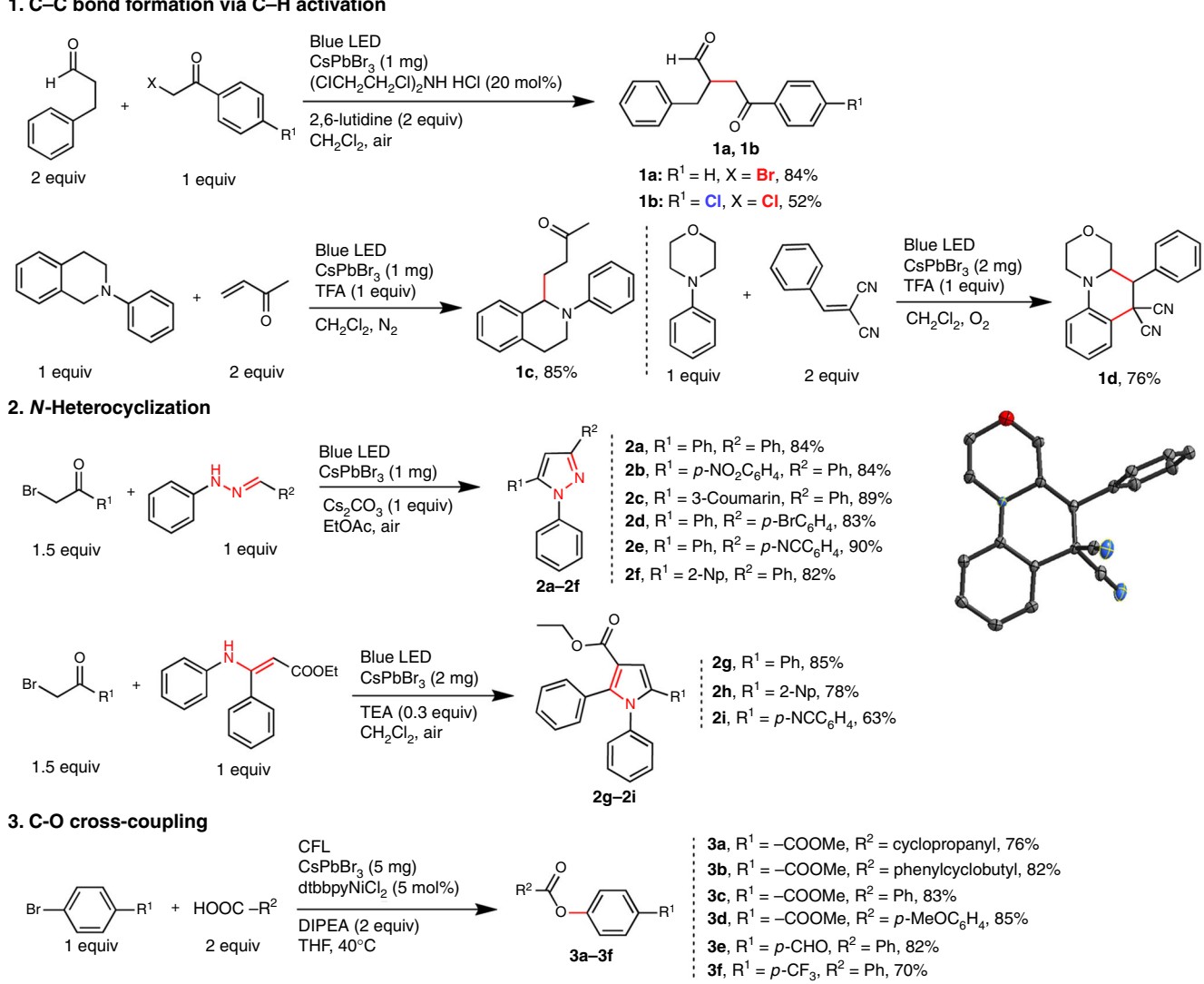

**Fig. 1** The library of C–C, C–N, and C–O bond formation reactions and respective yield. (Yields of **1a**, **1c**, **1d**, **2a**, **2g**, **3a** are the average yields of three reactions, see Supplementary Table 8; Inset: perspective view of **1d**'s single crystal structure with the thermal ellipsoids drawn at 50% probability level and the H atoms omitted for clarity.)

discrepancy is less obvious. In fact, in **2a**, perovskite is stable in a pre-dried non-halide solvent ethyl acetate (Supplementary Fig. 4). **2a** is produced in 86% yield with **P2** in 2 h, 87% using **P1** in 6 h (Supplementary Fig. 3). Overall, small size NCs, in general, promote a faster reaction rate, but not necessarily a higher yield unless presenting in a perovskite friendly reaction environment. Considering synthesis merits, large size NCs, in general, provide higher yield although a longer reaction time in a scale of 6 h or higher is required.

**Stability and reaction condition tolerance.** Pb-halide perovskites' photovoltaic performance perishes over moisture[28,29], impeding the wide commercial application of such materials as solar cells. The stability is quite distinct if perovskites are to be applied to organic synthesis in which more critical parameters may influence the stability of perovskites, i.e. solvent type, ions, acidity, *etc.*, and further manipulate the catalytic ability. Thus, these parameters are evaluated individually for a better understanding of perovskite photocatalysis. A quite strong stability of **P1** in organic solvents was indicated by no obvious PL changes of

$CsPbBr_3$ for several weeks in less polar organic solvents[13,17]. (Note that **P1** is not stable in polar solvents, i.e. acetone, acetonitrile, DMF, DMSO, Supplementary Fig. 4). However, **P2–P5** are less prominent and significant PL diminishing is observed. (Supplementary Fig. 2) Interestingly, under the irradiation of LED, PL blue-shift of **P1** in $CH_2Cl_2$ are observed in 24 h. (Fig. 2e) Such changes are significantly magnified on **P2–P5** as shown in Fig. 2e and Supplementary Fig. 5, absorption and PL blue-shift within in 1 h, whereas no obvious PL changes are observed in non-halide solvents. This observation may be attributed to a photoinduced fast halide exchange for $CsPbBr_3$ with $CH_2Cl_2$ as previously reported[16,30].

Next, we evaluate the ion effect in perovskites' photocatalysis. Perovskite is reported to sensitive to both inorganic cations and anions[31–34]. In our photocatalytic setup, co-catalyst $(ClCH_2CH_2)_2NH_2Cl$ in reaction **1a**, leads to an initial PL blue-shifted due to anion-exchange forming $CsPbBr_xCl_{3-x}$, confirmed by XRD (Fig. 2f). It is interesting to point out that co-formation of Br ion during reaction **1a**, may further exchange with the $CsPbBr_xCl_{3-x}$ and stabilize the perovskite NCs. Such

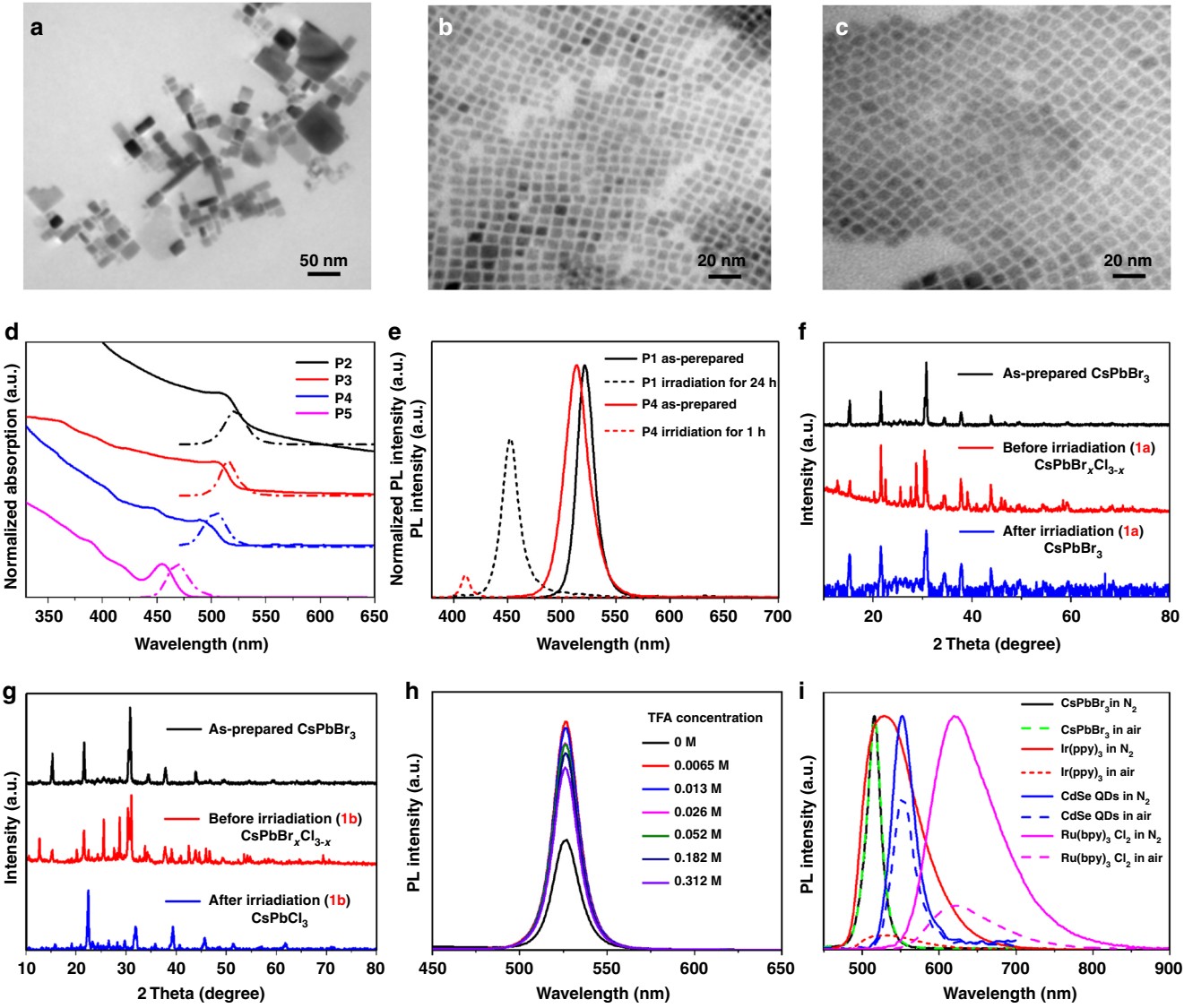

**Fig. 2** Characterization and spectroscopy studies of photocatalysts. **a** TEM of CsPbBr$_3$ **P1**; **b P3**; **c P4**; **d** UV-vis and PL spectra of CsPbBr$_3$ **P2–P5**; **e** PL spectra for **P1** and **P4** in CH$_2$Cl$_2$ as prepared and after LED irradiation for 24 h and 1 h, respectively. **f** XRD of as-prepared CsPbBr$_3$ **P1**; isolated from the reaction **1a** before and after irradiation, respectively; **g** the corresponding XRD for reaction **1b**; **h** PL spectra of **P1** in THF with addition of TFA; **i** PL spectra of CsPbBr$_3$ NCs, Ir(ppy)$_3$, CdSe QDs and Ru(bpy)$_3$Cl$_2$ in air or N$_2$-saturated solutions. Source data are provided as a Source Data file

stabilization is evidenced by the after-reaction catalyst characterization in which XRD indicates that the isolated photocatalyst solid was corresponding to CsPbBr$_3$ and surprisingly, no peak has been assigned to CsPbCl$_3$ (Fig. 2f). This is probably because the co-formation Br ions are in chemical equivalency and its concentration is significantly higher than that of Cl. Therefore, a Br compensated and stabilized CsPbBr$_3$ **P1** photocatalyst system is thus observed. (Supplementary Fig. 6) In contrast, reaction **1b** employing Cl-substrates leads to a fully-exchanged CsPbCl$_3$ after reaction (Fig. 2g). Overall, perovskite **P1** shows a much better stability during the reaction **1a**, in which the NCs can be isolated from the previous reaction mixture via centrifuging and then re-suspended for a new reaction under identical conditions for at least four cycles with slightly PL blue-shift, whereas small NCs **P4**'s recycling ability is limited (Supplementary Fig. 3). As comparison, when free halide anions are absent, for example in reaction **2a** in EtOAc solution, the overall stabilities for **P1** and **P4** are enhanced and result in an

improved recyclability in such perovskite friendly environment (Supplementary Figs. 6 and 7).

Acidity or free protons in perovskite reaction mixture may play a role in organic synthesis. For instance, carboxylic acids such as propionic acid, benzoic acid or trifluoroacetic acid (TFA), were employed as the co-catalyst (**1c** and **1d**) or as a substrate (**3a–3f**). Thus, we first measured the PL for perovskite NCs with different acids to elucidate the tolerance of acidic conditions. Interestingly, as shown in Fig. 2h and Supplementary Figs. 8 and 9, a PL enhancement of **P1** was observed upon the addition of benzoic acid, propionic acid, and also TFA (see Supplementary Movie 1). This is corroborated with previously observed PL enhancement using thiophenol[16], phosphoric acid[35] etc. The PL enhancements are probably because carboxylic acid function as the capping ligand by the strong hydrogen bonding with surface halide ions[35] and may also account from a strong interaction between carboxylic acid and Pb atoms, indicated by Tan et al.[36]. Acid binding with defects on perovskite may also lead to an enhanced

**Table 1 Comparison of photocatalysts for corresponding reactions in air or in oxygen**

| Photocatalyst[a] | Yield (%)[b] | | | | | TON (based on CsPbBr$_3$)[c] | | | | |
|---|---|---|---|---|---|---|---|---|---|---|
| | 1a | 1c | 1d | 2a | 3f | 1a | 1c | 1d | 2a | 3f |
| CsPbBr$_3$ **P1** | 84 | 85 | 76 | 84 | 70 | 9,100 | 830 | 280 | 380 | 33 |
| Ru(bpy)$_3$(PF$_6$)$_2$ | Trace | 60 | 25 | Trace | N.R. | – | 60 | 25 | – | – |
| Ir(ppy)$_3$ | 79 | N.R. | N.R. | 63 | 65 | 79 | – | – | 63 | 33 |
| CdSe QDs (525 nm) | Trace | Trace | N.R. | N.R. | N.R. | – | – | – | – | – |

[a]bpy 2,2'-bipyridine, ppy ortho-metalated 2-phenylpyridine
[b]average yield using for **P1**
[c]details in Supplementary Note 2

PL performance according to Zhu et al.[37]. The maximum PL was observed using TFA at a concentration of *ca.* 6.5–13 mM, more acid leads to a diminishing PL probably because large number of protons may start to initiate a deactivation process. Interestingly, such optimized TFA concentration also leads to a maximum product yield of **1c** and **1d** as shown in Supplementary Tables 3 and 4, indicating that a high PL of the photocatalyst may increase the catalytic conversion. Therefore, non-halide organic acid may not only stabilize the perovskite NCs, but also may increase the overall catalytic efficiency for respective reactions.

**Key catalytic parameter comparison with other photocatalyst.** Air-tolerance is important for the practical end-use of chemical synthesis. One distinct advantage of our colloidal system is that the organic reactions observed here occur without the need for N$_2$-sparging. In stark contrast, molecular photocatalyst[38] necessitates air-free reaction conditions. The key difference here is that the perovskite NCs likely undergo faster quenching from the organic substrates, while quenching from air is negligible. (Fig. 2i and Supplementary Fig. 10) The reverse is true for most cases of molecular catalysts – quenching is substantial an., O$_2$ quenching is substantial and competitive with the catalytic reactions, leading to poor catalytic results. Hence, yields of reaction **1**[6,39–41], **2**[42,43], and **3**[44] in air with perovskite are significantly higher than with others. (Table 1, Supplementary Tables 1–7) For instance, **1a** were obtained in 85% yield in air using perovskite, but only resulted in trace amount with Ru(bpy)$_3^{2+}$. These results suggested that perovskite may exhibit a broad tolerance, particularly towards air.

Catalytic turnover number (TON) is compared and listed in Table 1. Heterogeneous catalyst, i.e. 3.0 nm CdSe QDs were reported to optimally render a TON of 79,100 (based on QD's molecular weight Mw, 88,000 g mol$^{-1}$) in glove box[45]. However, in our condition under air, no yield (nor TON) of **1**, **2**, and **3** can be obtained using CdSe QDs. In addition to air-sensitivity, CdSe's performance was also dependent on size and capping ligands[45]. While changing capping ligand on perovskite plays little role in the yield as shown in Supplementary Tables 2–4. This is probably because the capping ligands (e.g., *n*-octylammonium) that stabilize perovskite colloids are reported to function as A site to the perovskite APbX$_3$ structure[31], hence no extra stabilization protocol is required using perovskite nanocrystal for photocatalysis. Using the method in CdSe QDs[45] to calculate TON, **P2** NCs (14 nm, based on Mw, 8,015,000 g mol$^{-1}$, **P1–P5** TON see Supplementary Table 9, calculation details see Supplementary Note 1) renders 2,565,000. Perovskites' heterogeneous catalytic ability is validated via regaining strong PL after recovering the catalyst via centrifuge after reaction (Supplementary Fig. 7). To compare TON with molecular catalysts, TON calculation based on mole of metal (independent of size, CsPbBr$_3$, 579.8 g mol$^{-1}$) was carried out instead. For instance, four cycles of the reactions

afford a TON, at least 9,100 for **1a** (Table 1, details see Supplementary Note 2). Overall, one or two orders of higher TONs under our condition are observed using perovskite than others, except reaction **3** in which TON may rely on both perovskite and Ni co-catalyst.

Higher activity of perovskite than other photocatalysts may account from the intrinsic photophysical properties on charge separation and transfer. For example, the perovskite NC's ultrafast interfacial electron and hole transfer dynamics has been revealed by Lian et al.[46]. First, negligible electron or hole trapping has been found in perovskite NCs, facilitating photoredox catalytic cycle. In the presence of organic substrates (as electron or hole acceptors in photoredox organic synthesis[2]), photon-induced excitons in perovskite can be efficiently dissociated and separated[46]. For instance the half-lives of electron transfer to an organic electron-acceptor is reported to be ~65 ps, while charge recombination rate is reported about ~2 orders slower. The hole transfer dynamics from perovskite to an organic substrate is also reported to be 20 times faster than its recombination[46]. Such observation is also corroborating with our previous reports on the ultra-slow recombination velocity of perovskite both in CsPbBr$_3$ and CSPbI$_3$ single crystals and films[8,18–20]. Overall, the lack of electron and hole traps and fast interfacial electron transfer and hole transfer rates are imperative that may enable highly efficient perovskite induced photocatalysis. In fact, the superior performance is not surprising given that when employed in photovoltaics, the Pb-halide perovskites also perform much better (PCE, 24.2%) compared to transition metal-based dye-sensitized solar cells (11%), QD photovoltaics (12%) and organic photovoltaics (12%)[12].

**Mechanism.** Oxygen may be of an essential component in certain photoredox reactions. For instance, in Fig. 3a, radical addition product **1c** is achieved in nitrogen atmosphere while in a similar setup, air or oxygen atmosphere produces a ring-closure **1d** (crystal structure provided in Fig. 1). Oxygen is found to be the key reagent as the hydrogen atom acceptor that further induced the C–H activation on phenyl rings[39,40]. As shown in Fig. 3, the reaction mechanisms are proposed in which the key radical intermediates have been investigated. Upon Stern–Volmer studies (Supplementary Figs. 11–17), perovskite PL quenching by **1d-A** was observed ($k_q = 3.6 \times 10^8$ M$^{-1}$ s$^{-1}$, details see Supplementary Fig. 12 and Supplementary Note 3) and resulted in **1d-B** radical in the presence of oxygen. Intermediate **1d-B** and **1d-C** have been verified via radical trapping experiment employing 2,2,6,6-tetramethyl-1-piperidinyloxy (TEMPO) as a radical scavenger, through LC-MS (Supplementary Figs. 19 and 20). In the absence of oxygen, radical **1c-B** is also confirmed by TEMPO-trapped product (Supplementary Fig. 18) and further verified by the self-coupling **1c-C** via $^1$H NMR. It is worth mentioning that the

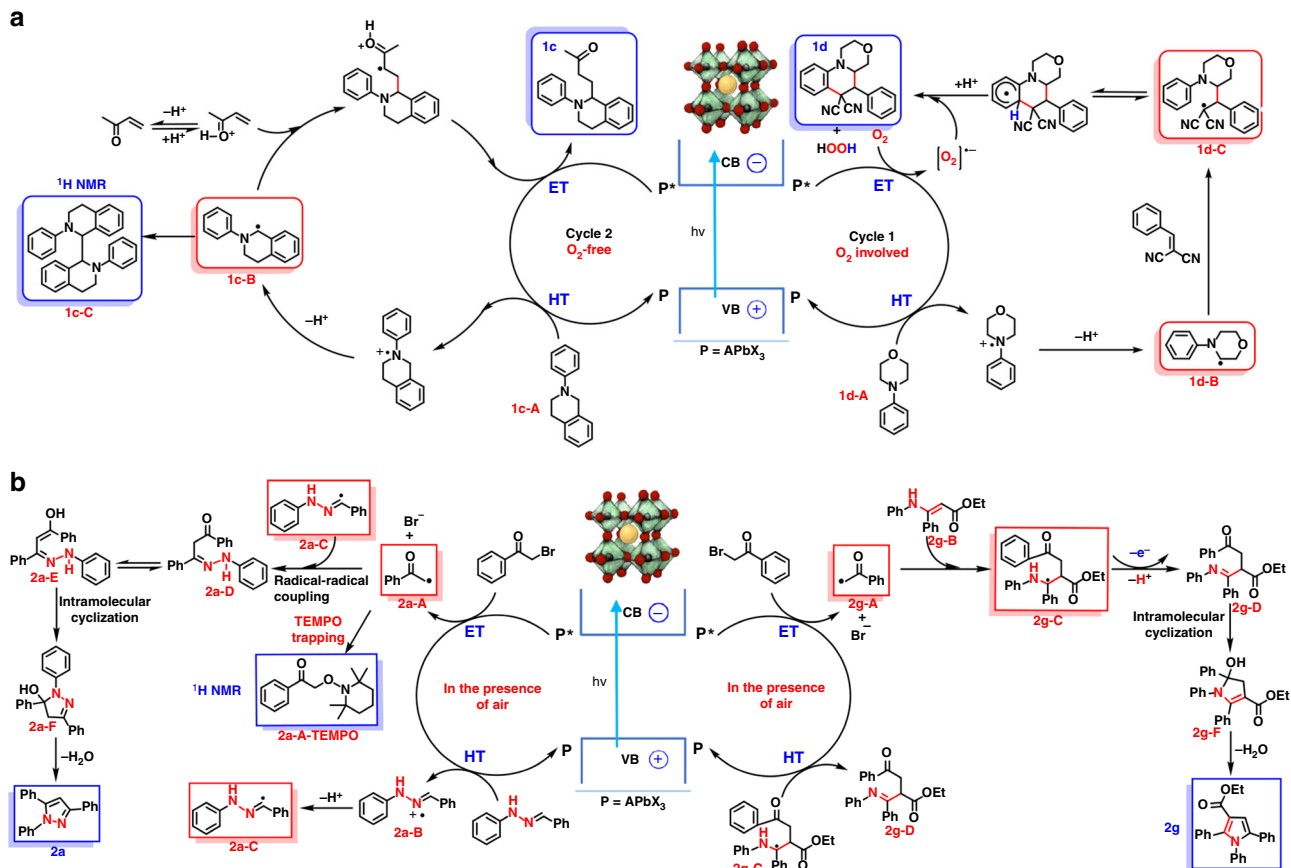

**Fig. 3** Mechanisms. **a** Proposed mechanisms for the synthesis of **1c** and **1d**; **b 2a** and **2g**. (Blue square: isolated and characterized by [1]H-NMR; Red square: trapped and detected by LC-MS (Supplementary Figs. 18–23); HT = hole-transfer; ET = electron-transfer)

presence of air leads to more **1c-C** formation and ultimately diminishes the yield of **1c**.

Figure 3b shows the proposed mechanism of C–N formations, in which both oxidative (ET, **2a-A**) and reductive quenching product (HT, **2a-B**) in reaction **2a** have been trapped by TEMPO (either observed via [1]H NMR or LCMS), indicating a strong charge separation and transfer ability induced by perovskite. This pathway is similar to our previous mechanism exploration in α-alkylation of aldehydes[17]. Radical coupling between **2a-A** and **2a-C** leads to the intermediate of **2a-D**. Then C–N formation via intramolecular cyclization and a final dehydration leads to the pyrazole product **2a**. In contrast, the radical formation from **2g-B** via direct HT has not been observed, instead **2g-C** was verified via radical-trapping, likely demonstrating a different mechanism of pyrrole formation as shown in Fig. 3b.

To further elucidate the reaction mechanism, electrochemical studies were conducted. (Supplementary Figs. 24–31) According to the comparison between redox potentials of the key substrates and the band energy of perovskite, the respective driving force is listed in Fig. 4. Driving force for HT in reaction **1c**, **1d** and **2a** is observed among ~0.1 to 0.3 eV, consistent with the Stern–Volmer quenching results (Supplementary Figs. 11–17) as well as the mechanistically verified intermediates in Fig. 3. However, **2g-B** disfavors HT due to a more positive oxidation potential ($E_{ox}$, 1.42 V vs SCE), corroborating with the previous observation that direct radical forming from **2g-B** is difficult, unlike reaction **2a** pathway. Moreover, driving force for ET is also listed from ~0.2 to 0.5 eV, confirming our discussion on ET in Fig. 3. However, noticeable exception, 2,4′-dichloroacetophenone, though presenting a more negative reduction potential ($E_{red}$, −1.47 V vs SCE),

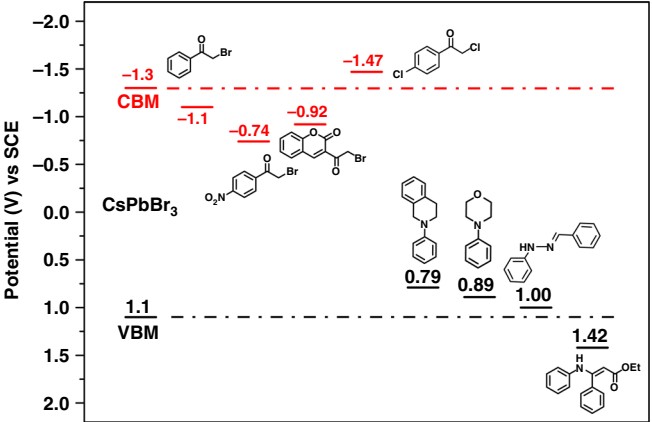

**Fig. 4** Band energy of CsPbBr$_3$ vs the redox potentials of substrates. Source data are provided as a Source Data file

still reacts to form respective pyrrole. We postulate that in-situ band-tuning of perovskite may play a role here and is discussed below.

**Unique band-tuning of perovskite.** As discussed above, the perovskite NCs **P1** are too large to be in the quantum-confinement regime and the majority of the NCs within the ensemble are larger than the Bohr radius. Thus, the band energy of our photocatalyst, analogs to excited state redox potentials, $E^*$

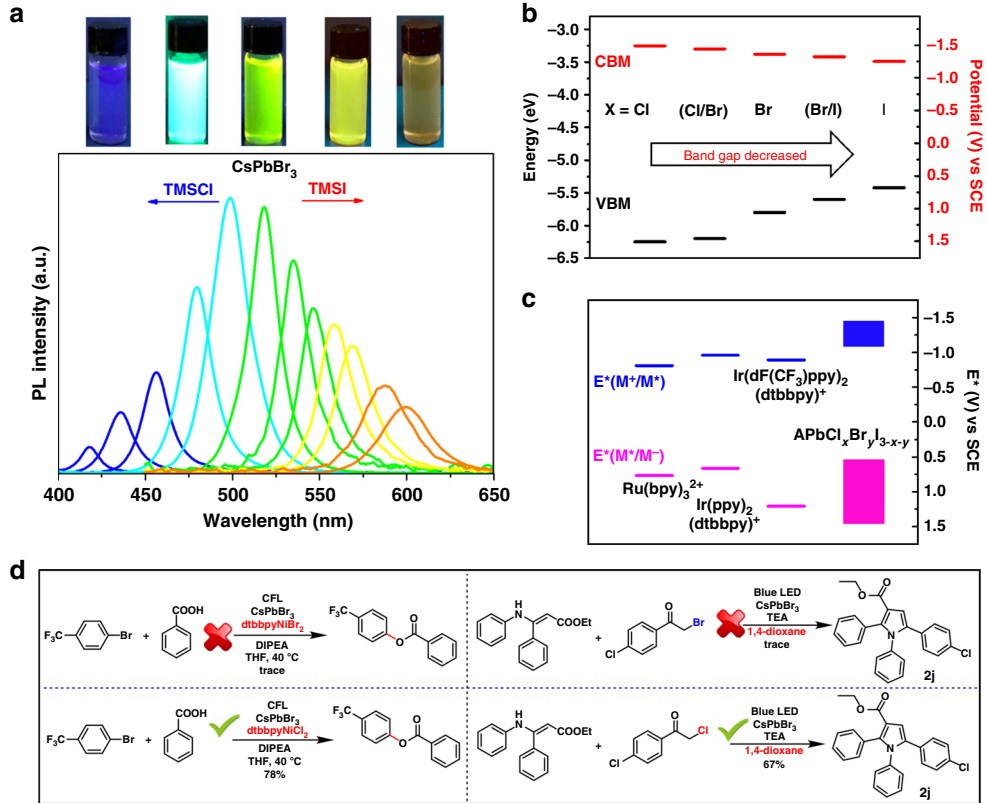

**Fig. 5** Band-tuning of perovskite. **a** The PL spectra of colloidal $CsPbBr_3$ in dichloromethane via band tuning with trimethylsilyl chloride or iodide and their representative images under UV lamp (top). **b** Bandedges of $APbCl_xBr_yI_{3-x-y}$. **c** Excited-state potential ($E^*$) range of $APbCl_xBr_yI_{3-x-y}$ comparing with noble transition-metal catalysts. **d** Two successful reaction examples with perovskite band-tuning. Source data are provided as a Source Data file

in molecular catalyst, is determined by the bulk bandedges. Bandedge-tuning is achievable by simply mixing of different ratio of halides[32,33,47]. We also observed that in-situ ion exchange using **P1** results in band-tuning (Fig. 5a). In theory, as shown in Fig. 5b, c the bandedges of perovskite after tuning covers most of the $E^*$ of the known Ru or Ir molecular photocatalysts.

The band-tuning is of critical importance for a photocatalyst to activate different types of substrates. For example, C–O formation reaction **3** is also proposed and shown in Supplementary Fig. 32 similar to previously reported mechanism[44]. It is reported that energy transfer from triplet excited state of Ir photocatalyst is the key for Ni complex activation thus resulting in an efficient reductive elimination for C–O bond formation[44]. Triplet energy ($E_T$) exploration from $Ir(ppy)_3$ derivatives via modifying the substitution group on ppy ligand demonstrated that a higher correlation between $E_T$ and the production yield. Specifically, a higher $E_T$ results in a higher yield. As shown in Fig. 5d, in our perovskite system, **3f** is produced in trace amount if $CsPbBr_3$ is employed with $dtbbpyNiBr_2$ co-catalyst, comparing to 78% with $dtbbpyNiCl_2$. While in Ir photocatalysis, the different halides on Ni co-catalyst only play a marginal effect[44]. We suspect that an in-situ ion-exchange from $NiCl_2$ may result in a blue shift of perovskite, similar to the increasing $E_T$ in Ir system, thus leading to a significantly higher yield of **3f** using co-catalyst $dtbbpyNiCl_2$. To further confirm such hypothesis, we have conducted a systematic band-tuning experiment to demonstrate the correlation between the bandedges and the yield of **3f**. In a typical experiment, perovskite $CsPbBr_3$ is employed with $NiBr_2$ co-catalyst, but tuned using a reported agent, i.e. trimethylsilyl chloride (TMSCl)[34]. We find that shifting the bandgap to higher values, by mixing with chloride to form $CsPbCl_xBr_{3-x}$, increases

the yield of **3f**, similar to elevate $E_T$ in Ir system. However, more Cl component is not always beneficial for this type of reaction. As shown in Fig. 5a, PL intensity is significantly lower when Cl is incorporated into perovskite. Higher bandgaps (shorter PL peak wavelengths) resulted in a lower yield, and is likely tied to the lower PL quantum efficiency that indicates a competitive carrier trapping mechanism[32]. Overall, a maximum yield of 85% was obtained when the PL peak corresponds to 498 nm (Supplementary Table 7). This observation illustrates that the intentional band-tuning of perovskite NCs may activate previously non-reactive substrates.

Furthermore, band-tuning may also result in an absolute discrepancy in photo-activation. It is widely accepted that the C–Cl bond are stronger than C–Br and hence harder to activate[48]. Surprisingly, in reaction **2j** in Fig. 5d, α-chloroketone is observed to react to form **2j** in dioxane in a yield of 67% while α-bromoketone is almost non-reactive at all. We assume that the band energy of $CsPbBr_3$ is not adequate to activate either Cl or Br-substrates in dioxane. However, ion-exchange may not only occur between $CsPbBr_3$ and $CH_2Cl_2$ as previously reported[16,30], but may also between $CsPbBr_3$ and suitable organic Cl-substrates. Interestingly, $CsPbCl_3$ was confirmed by XRD after reaction. (Supplementary Fig. 33) Cl-substrate is the only Cl source and hence is activated in this type of reaction. Hence CB of photocatalyst is thus moved higher, simultaneously the reduction potential of the substrate moves in a reverse direction (Fig. 5b), overall making the ET possible and finally resulting in the pyrrole formation. This is also corroborating with the observation that, if reaction **2j** was conducted in $CH_2Cl_2$, $CsPbBr_3$ NCs may first initiate ion-exchange with the solvent hence shift band energy, and finally catalyze both Br and Cl-substrates towards **2j**

formation. This result demonstrates that in-situ band-tuning of perovskite NCs may provide unexpected activity towards previously unachievable substrates.

## Discussion

In summary, a general acceptance of perovskite nanocrystals for organic reactions has been demonstrated. C–C bond formations via C–H activation, C–N, and C–O formations via N-heterocyclizations and aryl-esterifications can be achieved with moderate to high yields. Large size perovskites NCs with band energy determined by bulk $CsPbX_3$, in general provided higher yield for above reactions than perovskites quantum dots, probably due to a stability concern. A detailed stability study of perovskites regarding solvent type, ions, acidity has been explored. We also demonstrate that oxygen quenching of perovskite is less efficient. Therefore, perovskite colloids are much more active than most of those developed catalysts in air. Such tolerance may render perovskite a much broader activation for organic synthesis, particularly towards air. Mechanistic investigation further proves perovskites' excellent property towards photo-induced charge separation and transfer. Moreover, easy and wide bandedge tuning of the Pb-halide perovskites provides for achieving a key challenge in activating a broader range of organic substrates that require vastly different energy levels. Intentional or in-situ band-tuning experiment of $CsPbBr_3$ NCs exhibits that previously unachievable reactions, i.e. 2j, 3f, can be re-activate via a simple anion-exchange protocol. We envision that the photophysical knowledge that demonstrated in perovskite solar cell may be transformative for photocatalytic organic reactions. The broader application of this air-tolerant, cost-effective, easily-prepared, highly-active and band-tunable lead halide perovskites may be of a revolutionary breakthrough in the photocatalysis of organic reactions.

## Methods

**General considerations**. Commercial reagents were purchased from Sigma Aldrich and TCI America. Additionally, aldehydes were distilled prior to use. Tetrahydrofuran was distilled under $N_2$ over sodium benzophenoneketyl. All other solvents were purified by passage through columns of activated alumina. Two batches of CdSe Quantum Dots with nanoparticle concentration of 50 μmol/L in hexane with emission peak at 525 nm (particle size 2.8 nm) and 550 nm (particle size 3.5 nm) were purchased from Strem Chemicals. Silica gels (230–400 mesh) used for chromatography were purchased from Sorbent Technology. $^1$H NMR and $^{13}$C NMR spectra were recorded in $CDCl_3$ on Bruker spectrometers at 400 or 500 ($^1$H NMR) and 100 or 125 MHz ($^{13}$C NMR). All shifts are reported in parts per million (ppm) relative to residual $CHCl_3$ peak (7.27 and 77.2 ppm, $^1$H NMR and $^{13}$C NMR, respectively). All coupling constants (J) are reported in hertz (Hz). Abbreviations are: s, singlet; d, doublet; t, triplet; q, quartet; brs, broad singlet. High-resolution mass spectra (HRMS) were measured on a 7T Bruker Daltonics FT-MS instrument. LC-MS spectra were measured on a Thermo Finnigan LTQ MS/MS with Agilent 1100 LC front end for MS with binary pump. TLC analysis was carried out on glass plates coated with silica gel 60 F254, 0.2 mm thickness. The plates were visualized using a 254 nm ultraviolet lamp or aqueous potassium permanganate solutions. $^1$H NMR data are given for all compounds for characterization purposes. $^1$H NMR, $^{13}$C NMR, and HRMS data are given for all new compounds.

A Shimadzu UV-2501 spectrophotometer was used to record the UV-vis absorption spectra in different solvents. A Horiba Fluoro-Max 4 fluorometer/phosphorometer was utilized to measure the steady-state emission spectra. Hitachi H-7500 transmission electron microscope was utilized to measure the TEM images. Philips Empyrean X-Ray Diffractometer was used to measure powder XRD.

**Cyclic voltammetry measurement**. The electrochemical experiments were carried out using a CHI 600E electrochemistry workstation (CHI, USA). A three-electrode cell was used with a Pt disc electrode as the working electrode, a Pt wire as the counter electrode and an Ag/AgCl electrode (Ag in 0.1 M $AgNO_3$ solution, from Sigma-Aldrich) as the reference electrode. Tetrabutylammonium hexafluorophosphate (0.1 M) was used as the supporting electrolyte. The potential values obtained in reference to Ag/AgCl were converted to the saturated calomel electrode (SCE) in order to directly compare with literature. All solutions were purged with $N_2$ for 20 min before experiments.

**X-ray crystallographic analysis**. Single crystals of 1d were obtained by slow diffusion of diethyl ether into dilute dichloromethane solution. A suitable crystal of 1d (CCDC 1889861) was selected and collected on a Bruker Apex Duo diffractometer with an Apex 2 CCD detector (Bruker, Madison, WI, USA) at T = 273 K, respectively. Mo radiation was used. The structure was processed with an Apex 2 v2010.9-1 software package (SAINT v.7.68A, XSHELL v.6.3.1)[49,50]. A direct method was used to solve the structure after multi-scan absorption corrections. Details of data collection and refinement are given in Supplementary Tables 10–12.

**Synthesis of perovskite CsPbBr$_3$ P1**. $CsPbBr_3$ P1 NCs were synthesized by the modification of the method reported[17,24]. First, two precursor solutions are prepared in advance: 2.0 mmol CsBr dissolved in 2.0 mL $H_2O$ and 2.0 mmol $PbBr_2$ dissolved in 3 mL DMF, respectively. Then, to a vigorously stirring mixture of 500 mL hexane, 8 mL oleic acid and 1.5 mL n-octylamine, the $PbBr_2$ DMF solution and CsBr solution are added dropwise. Along with mixing, an emulsion forms and the solution color turns from clear to slightly white. After that, acetone (400 mL) is added to break-up the emulsion. The $CsPbBr_3$ NCs are isolated by centrifugation at 2000 rpm for 2 min to discard large particles, and then 7000 rpm for 10 min to afford $CsPbBr_3$ P1.

**Synthesis of CsPbBr$_3$ P1-oleyamine**. Use the very similar method with the synthesis of Perovskite $CsPbBr_3$ P1, except the using of oleyamine instead of octylamine.

**Colloidal CsPbBr$_3$ P2–P5**. $CsPbBr_3$ P2–P5 NCs were synthesized according to the previously reported method[51]. First, $Cs_2CO_3$ (0.814 g) was loaded into 100 mL 3-neck flask along with octadecene (40 mL) and oleic acid (2.5 mL, OA), dried for 1 h at 120 °C, and then heated under $N_2$ to 150 °C until all $Cs_2CO_3$ reacted with OA. Then, 5 mL ODE and $PbBr_2$ (0.069 g, 0.188 mmol) are loaded into 25 mL 3-neck flask and dried under vacuum for 1 h at 120 °C. Dried oleylamine (0.5 mL) and dried OA (0.5 mL) were injected at 120 °C under $N_2$. After complete solubilization of $PbBr_2$, the temperature was raised to a desired value, and the prepared Cs-oleate solution (0.4 mL, 0.125 M in ODE) was quickly injected and, 5–10 s later, the reaction mixture was cooled by immersion in an ice-water bath. After centrifugation at 5000 rpm for 5 min to discard the precipitates, a bright yellow-green colloidal solution was obtained. The synthesized $CsPbBr_3$ are precipitated by adding 6 mL n-butanol and then centrifuged at 12000 rpm.

**Synthesis of CsPbBr$_{3-y}$X$_y$ (X = Cl, I)**. First, the colloidal $CsPbBr_3$ are prepared in $CH_2Cl_2$. Subsequently, the different volumes of trimethylsilyl chloride (TMSCl) or trimethylsilyl iodide (TMSI) DCM solution is dropped into the $CsPbBr_3$ solution until the desired emission peak position is achieved.

**Photocatalytic organic synthesis procedure**. In a typical synthesis, for instance 1a and 1b, to a 4 mL vial, $CsPbBr_3$ NCs P1 (1.0 mg), the corresponding bromide or chloride (0.5 mmol, 1.0 equiv.), 3-phenylpropanal (1.0 mmol, 2.0 equiv.), 2,6-lutidine (1.0 mmol, 2.0 equiv.), bis(2-chloroethyl)amine hydrochloride (0.1 mmol, 0.2 equiv.), and 2 mL $CH_2Cl_2$ were added and then stirred under the irradiation with a 12 W 455 nm Blue LED lamp, distance ~8 cm. After 5~12 h, the mixture was poured into water, and extracted with $CH_2Cl_2$ (3 × 10 mL). The combined organic layers were washed with water, dried over $Na_2SO_4$ and concentrated in vacuo. The crude product was purified by column chromatography (silica gel, Hexane/EtOAc = 10:1) to afford 1a or 1b. For $^1$H NMR and $^{13}$C NMR spectra of all compounds see Supplementary Figs. 34–58. Full experimental details can be found in the Supplementary Methods.

## Data availability

The authors declare that the data supporting the findings of this study are available within the paper and its Supplementary Information file. The X-ray crystallographic coordinates for structures of 1d has been deposited at the Cambridge Crystallographic Data Centre (CCDC) under deposition number CCDC 1889861. The data can be obtained free of charge from the Cambridge Crystallographic Data Centre via http://www.ccdc.cam.ac.uk/data_request/cif. The source data underlying Figs. 2, 4 and 5, and Supplementary Figs. 2, 3, 5, 7–17, 24–31 and 33 are provided as a Source Data file.

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

## Acknowledgements

This research is partially supported (C–C bond formation) by NSF under Chemical Catalysis program, award 1851747 to Y.Y. Y.Y. also acknowledges the support as part (C–O bond formation) of the Center for Hybrid Organic Inorganic Semiconductors for Energy (CHOISE) an Energy Frontier Research Center funded by the Office of Science, Office of Basic Energy Sciences within the US Department of Energy. We also thank N. Yamamoto for help in the X-ray crystallographic measurement.

## Author contributions

Y.Y. conceived the original idea and led the project; X.Z., Y.L., J.S.M., Y.S., D.Z., and Y.Y. carried out the perovskites synthesis, characterization, and catalysis investigation; Y.Y. wrote the manuscript with inputs and discussions from all authors.

## Additional information

**Competing interests:** The authors declare the following competing interests: A provisional patent application has been filed on the perovskite catalysts and their use in photocatalytic organic synthesis.

