## [Peer Review File · Nature Communications]

Reviewers' comments:

Reviewer #1 (Remarks to the Author):

This manuscript describes using lead halide perovskites as photocatalysts for organic synthesis. The authors have made substantial improvement from their previous submission based on all the review comments. The authors have conducted very careful research and I really like some of their study. For example, the authors compared the catalytic activity and the size of different perovskite colloids, and found that perovskite colloid they made has a wide range of size distribution (2-100 nm), which leads to a higher stability. It is interesting to note that "small size NCs in general promote a faster reaction rate, but not necessarily a higher yield unless presenting in a perovskite friendly reaction environment". They also provided the data for comparison of perovskite nanoparticles with other molecular photocatalysts, and found perovskite showed better performance to tolerate oxygen environment. The most intriguing discover to me is that the authors found organo-chloride substrates worked effectively than organo-bromides in perovskite catalyzed photo-transformations. I have to say this is a fabulous discovery as in general organo-bromides are better electrophiles than organo-chlorides in organic synthesis. The in situ tuning of the band gap of perovskite enables the opposite reactivity, which can be very interesting and potentially useful. Overall, this is a highly timely topic and could be very stimulating to the research field of photocatalyst. I would be happy to recommend its publication in Nature Communication with minor revisions:

1. The reference background description needs to be more precise. For instance, the authors only claimed "benzenethiol dimerization" in ref. 16, however, C-H and C-Cl activation using perovskite as photocatalysts have also realized in this reference. Even though there is a little overlap with this manuscript, I don't think it will be harmful to state them out. The novelty and systematic study in this manuscript is at another level.
2. The authors describes that P1 is stable in less polar organic solvents. How about polar solvents? It is instructive to know the solvent scope that perovskites can tolerate.
3. Is there any possibility that the enhancement of PL intensity of perovskite NCs with acids is caused by binding the defects on perovskites?

Reviewer #2 (Remarks to the Author):

In general, the revised MS has been improved compared to the previos version, however, it seems that the author in the mainwhile have published the alkylation of aldehydes in a recent JACS publication (J. Am. Chem. Soc. 141, 733-738 (2019)). This also triggered me to search for other recent work on perovskites used in organic synthesis and it seems that several publications are missing from the state-of-the-art discussion (ACS Energy Lett., 2019, 4 (1), pp 203–208 and ACS Energy Lett., 2018, 3 (4), pp 755–759). While several issues have been resolved in this revised manuscript, there are still some major scientific and textual issues in this version that should at least be addressed before publication.

1. This paper gives the reader a good view on the organic reactions that perovskite can perform, however no details about the mechanism is provided. I strongly suggest that the author should give us more insight in the reaction cycle, intermediates, active species...

2. In the second part of this paper, the author try to reveal the size effect of perovskite on the reaction. However, only two sizes, 6nm and 9nm, were tested. From the principle of statistics, it is difficult to obtain any information from this limited comparison.
3. The perovskite is instability in CH₂Cl₂, as the author said it can be attributed to the ion exchange, XRD or XPS is a better method to prove it. "whereas no obvious PL changes are observed in non-halide solvents (Fig. S6)." but I do not see the relevant PL spectrum in Figure S6. In Fig 2f and 2g, the author try to use XRD to prove the ion change during the photocatalytic reaction, but the logic is too complex. It would be more intuitive if the author make a comparison on CsPbBr₃ between before and after reaction.
4. The author use acid to enhance the PL intensity and obtain a conclusion that "non-halide organic acid may not only stabilize the perovskite NCs, but also may increase the overall catalytic efficiency for respective reactions (1a, 1d and 1e)". Can the author use photocatalytic reactions(1a, 1d and 1e) to prove this conclusion?
5. From the stability test in SI, it seems like the stability of perovskite during photo reaction is too poor. P2 seems like loss the activity suddenly. what is happening at that point?
6. I am confused by the author's statement about the ligands. CdSe QD is stabilized by oleic acid, the author's sample still need ligand to stabilize, n-octylammonium, I do not see any advantage about this point.
7. In the last part, replacing Br with Cl and I to tune the bandgap has been described in many publications. Some contents are similar to the author's JACS paper.

Response to Reviewer 1

Reviewer #1

Comments:

This manuscript describes using lead halide perovskites as photocatalysts for organic synthesis. The authors have made substantial improvement from their previous submission based on all the review comments. The authors have conducted very careful research and I really like some of their study. For example, the authors compared the catalytic activity and the size of different perovskite colloids, and found that perovskite colloid they made has a wide range of size distribution (2-100 nm), which leads to a higher stability. It is interesting to note that “small size NCs in general promote a faster reaction rate, but not necessarily a higher yield unless presenting in a perovskite friendly reaction environment”. They also provided the data for comparison of perovskite nanoparticles with other molecular photocatalysts, and found perovskite showed better performance to tolerate oxygen environment. The most intriguing discover to me is that the authors found organo-chloride substrates worked effectively than organo-bromides in perovskite catalyzed photo-transformations. I have to say this is a fabulous discovery as in general organo-bromides are better electrophiles than organo-chlorides in organic synthesis. The in situ tuning of the band gap of perovskite enables the opposite reactivity, which can be very interesting and potentially useful. Overall, this is a highly timely topic and could be very stimulating to the research field of photocatalyst. I would be happy to recommend its publication in Nature Communication with minor revisions:

Reply: We're glad to see this reviewer's support on the publication of our work. We really appreciate this reviewer's positive comments and we would also be happy to revise according to this reviewer's suggestions.

1. The reference background description needs to be more precise. For instance, the authors only claimed “benzenethiol dimerization” in ref. 16, however, C-H and C-Cl activation using perovskite as photocatalysts have also realized in this reference. Even though there is a little overlap with this manuscript, I don't think it will be harmful to state them out. The novelty and systematic study in this manuscript is at another level.

Reply: Thank you very much for this valuable comment. We have revised the manuscript accordingly and added C-H activation towards C-P formation in the revised manuscript. In addition we also added the C-Cl activation from this reference via dichloromethane (solvent) based halide exchange with perovskite nanocrystals in the updated manuscript. We really appreciate this reviewer's comment on our work that “*The novelty and systematic study in this manuscript is at another level.*”

2. The authors describes that P1 is stable in less polar organic solvents. How about polar solvents? It is instructive to know the solvent scope that perovskites can tolerate.

Reply: Thanks for this comment. And we are sorry for missing the detail on this information. CsPbBr₃ **P1** NCs are quite stable in less polar solvents, such as Hexane, Toluene, 1,4-dioxane, EtOAc *etc.*; However, **P1** NCs are less stable in polar solvents, such as Acetone, Acetonitrile, DMF, DMSO *etc.* We have added the related detailed discussion and information both in the updated manuscript and in updated Figure S6.

Fig. S6. (a) The photograph for CsPbBr₃ **P1** disperse in less polar solvents: Hexane, Toluene, 1,4-dioxane, ethyl acetate; polar solvents: Acetone, acetonitrile, DMF, and DMSO at ambient light (top) and 365 nm UV light (bottom)

3. Is there any possibility that the enhancement of PL intensity of perovskite NCs with acids is caused by binding the defects on perovskites?

Reply: We indeed appreciate this reviewer's comment on the binding of defects on perovskite. We noticed that bind of acid on perovskite may result in a better PL as well as elevated solar cell performance, according to reference *Energy Environ. Sci.*, 2018,11, 3480-3490. In this reference, Lewis acid is added into perovskite film to significantly suppress defects through a synergistic effect, hence a better PL performance is indeed observed in this work. And finally such process leads to a highly efficient and stable perovskite solar cell. We therefore agree with this reviewer that there is a possibility that that the enhancement of PL intensity of perovskite NCs with acids is caused by binding the defects on perovskites. Related discussion is added in the manuscript.

In these regards, we really appreciate this reviewer's excellent comments to indeed enhance the quality of this work.

Response to Reviewer 2

Reviewer #2:

In general, the revised MS has been improved compared to the previous version, however, it seems that the author in the meanwhile have published the alkylation of aldehydes in a recent JACS publication (J. Am. Chem. Soc. 141, 733-738 (2019)). This also triggered me to search for other recent work on perovskites used in organic synthesis and it seems that several publications are missing from the state-of-the-art discussion (ACS Energy Lett., 2019, 4 (1), pp 203–208 and ACS Energy Lett., 2018, 3 (4), pp 755–759). While several issues have been resolved in this revised manuscript, there are still some major scientific and textual issues in this version that should at least be addressed before publication.

Reply: We indeed thank this reviewer very much for the valuable comments and suggestions. These comments and suggestions indeed help us to revise and improve the quality of this work. Therefore, we have added the related references and made revisions accordingly.

Regarding our recent published JACS, it is our initial exploration of perovskite towards photocatalytic α -alkylation of aldehydes. It is a first attempt that a very simple C-C bond formation reaction can be achieved using perovskite in high yield. However, as we discussed in the introduction, more general acceptance of perovskite toward photoredox reactions that are of broader interests is still unknown. Particularly, the challenging synthesis towards much more useful organic synthesis, *i.e.* drug molecules: such as ring-closure C-C bond forming, *N*-heterocyclization, aromatic amination or aryl-esterification; or materials synthesis: polymerizations. We discuss the key parameters that are of interests from research fields both in perovskite materials and in organic chemistry. Such key parameters, include but not limited to, size effect, stability and tolerance, capping ligand effects, comparisons with known photocatalysts etc. More importantly, we have conducted experimental evidence to show that the bandgap-tuning via halides exchanging can be applied to activate different reactions and catalyze previously unachievable reactions. At the end of our recently published JACS paper, we briefly mentioned about the theoretical possibility that the band-tuning's effect on catalytic reaction may be achievable. However, so far as we know, band-tuning's **real effects** on photocatalytic reactions have not been experimentally discussed, nor explored. In this work, we are also devoted to unveiling the mystery of band-tuning of perovskite with regards to their photocatalytic ability. Therefore, we believe "the novelty and systematic study in this manuscript is at another level".

The comments mentioned from previous work on perovskite/NiO_x/TiO₂ solar cell photocatalysts published on *ACS Energy Lett* regarding C-H activations (C-H oxidation to form aldehyde or alcohol) and benzylic alcohol oxidation are very interesting and related to our work. And we are very sorry to miss these references

and are happy to add the related discussions. We also notice that these authors are very careful in terms of terminology. For example, such C-H activation or benzylic alcohol oxidation is not described as “organic synthesis” by these authors in these two *ACS Energy Letter* papers. In fact, these authors are very careful to describe the capability of such heterogeneous TiO₂/perovskite system for organic chemical activation. One critical point is that the product yield they provide for C-H activation, for example, is much less than 1%, some examples are less than 0.1%, NOT practical for organic synthesis. Another important point needs to mention is that oxidation of organics (or organic pollutants) using metal oxide mixed/doped TiO₂ (*i.e.* NiO_x in this paper) or visible light absorption dye-modified TiO₂ (*i.e.* perovskite in these papers) has been extremely common in the field of environmental science research area, but such oxidation can't be described or categorized as organic synthesis.

But we definitely agree with this reviewer that this type of work is relevant, and we would love to discuss and add these missing references. This type of discussion now is added in the revised manuscript. We do not think these papers will influence the fundamental insight of our work. But we greatly appreciate this reviewer's comments and would certainly love to address his/her comments one by one as demonstrated below.

1. This paper gives the reader a good view on the organic reactions that perovskite can perform, however no details about the mechanism is provided. I strongly suggest that the author should give us more insight in the reaction cycle, intermediates, active species...

Reply: Thank you very much for this suggestion. We have the proposed mechanism discussion in our first version when submitted to *Nat. Chem.* We removed it because we think the proposed mechanisms for many of the discussed reactions may demonstrate no obvious difference comparing to Iridium and Ruthenium noble-metal photocatalysts. However, thanks to this reviewer's comments, we therefore re-investigate each and every type of the reactions on their respected mechanism. And we do find interesting difference and we also discovered, observed or isolated some important intermediates to prove the proposed mechanism. During the mechanism exploration, we employed a radical scavenger TEMPO to trap the related intermediate to prove our proposed mechanism. Here in this revised manuscript, we have added the mechanism discussion/comparison of C-C bond formation via C-H activations in the absence of air or in the presence of air in Scheme 1a. And we also added the proposed mechanism of *N*-heterocyclization in Scheme 1c with experimentally observed intermediate. In scheme 1b, we have added the verified key radical intermediate via TEMPO trapped experiment. The rest of the proposed mechanisms, *i.e.* C-O, C-N bonds formations, or polymerizations *etc.* in supporting information as shown in Scheme S1-S2. The revision is listed below:

More importantly, oxygen may be of an essential component in certain photoredox organic reactions, in which perovskite may show a superior performance than oxygen sensitive photocatalysis as shown in Scheme 1. For instance, in C-C bond formations, **1d** is achieved in nitrogen atmosphere while in a similar setup, air or oxygen atmosphere produces a ring-closure **1e** (crystal structure provided). Oxygen is found to be the key reagent to be reduced as the hydrogen atom acceptor that further induced the C-H activation on phenyl rings.^{40, 41} As shown in Scheme 1, the reaction mechanisms are proposed in which the key radical intermediates have been investigated. For instance, perovskite PL quenching by **1e-A** was observed and resulted in **1e-B** radical in the presence of oxygen. Intermediate **1e-B** and **1e-C** have both been verified via radical trapping experiment employing 2,2,6,6-tetramethyl-1-piperidinyloxy (TEMPO), a radical scavenger, through LC-MS (Scheme 1b). In the absence of oxygen, radical **1d-B** has also been confirmed by TEMPO-trapped product and further verified with an isolated self-coupling compound **1d-C** via ¹H NMR (SI). It is worth mentioning that the presence of air leads to more **1d-C** formation and ultimately diminishes the yield of **1d**. In Scheme 1c, under air condition for **2a** and **2f** formations, substrate quenching of perovskite also leads to the formation of **2a-A** or **2f-A**, corroborating with **2a-B** molecule via TEMPO trapped experiment. Furthermore, the TEMPO trapped product for the radical intermediate **2f-B** has also been observed in LC-MS (Scheme 2b). Other proposed mechanisms for C-O, C-N formation and ATRP in our perovskite photocatalysis under air are shown in Scheme S1-S2. However, the competition between oxygen as a quencher for molecular photocatalyst and oxygen as a reactant, is essentially problematic and leading to low yield or no reaction of such oxygen-involving reactions (Table 1).

Scheme 1. (a) Proposed mechanisms for the synthesis of **1d** and **1e**; (b) LC-MS evidence for the TEMPO trapped products for **1e-B**, **1e-C**, **1d-B**, **2f-B**, respectively; (c) Proposed mechanisms for the synthesis of **2a** and **2f**.

Proposed Mechanism

Scheme S1. Proposed mechanisms for the C-N (left) and C-O (right) bond formation.

Scheme S2. Proposed mechanisms for ATRP.

Radical Trapping Experiments

The experimental procedure for trapping radicals with TEMPO

1) The trapping experiment for the synthesis of **1d**.

Scheme S3. TEMPO trapping experiment for **1d**.

To a 4 mL vial, CsPbBr₃ NCs (1.0 mg), 2-phenyl-1,2,3,4-tetrahydroisoquinoline (0.25 mmol, 1.0 equiv.), 3-buten-2-one (0.5 mmol, 2.0 equiv.), TEMPO (0.5 mmol, 2.0 equiv.), trifluoroacetic acid (0.05 mmol, 20 mol%), and 2 mL DCM, the mixture was bubbled with N₂ for 10 min and then stirred. After irradiation with Blue LED for 8h, trace amount of **1d** and **1d-C** was isolated, while 1d-B-TEMPO was detected by LC-MS. The crude self-coupling product **1d-C** was purified by column chromatography to afford a mixture of diastereoisomers with 1.28:1 dr. ¹H NMR (500 MHz, CDCl₃) δ 7.39-6.86 (m, 16H, ArH (major+minor)), 6.79 (t, 2H, ArH (minor)), 6.75 (t, 2H, ArH (major)), 5.37 (s, 2H, CHCH, major), 5.34 (s, 2H, CHCH, minor), 3.56-3.52 (m, 2H, NCH₂CH₂, minor), 3.42-3.25 (m, 4H, NCH₂CH₂, major+minor), 2.87-2.82 (m, 2H, PhCH₂CH₂, major), 2.69-2.59 (m, 2H, PhCH₂CH₂, major+minor), 2.08-2.02 (m, 2H, PhCH₂CH₂, minor). ESI-MS m/z calcd for C₃₀H₂₉N₂⁺ ([M+H]⁺) 417.2331, found: 417.2335.

2) The trapping experiment for the synthesis of **1e**.

Scheme S4. TEMPO trapping experiment for **1e**.

To an 8 mL vial, CsPbBr₃ NCs (1.0 mg), 4-phenylmorpholine (0.25 mmol, 1.0 equiv.), 2-benzylidenemalononitrile (0.5 mmol, 2.0 equiv.), trifluoroacetic acid (0.05 mmol, 20 mol%), and 1 mL DCM were added, the mixture was bubbled with O₂ for 10 min and then stirred. After irradiation with Blue LED for 8h, trace amount of **1e-B-TEMPO** and **1e-c-TEMPO** was detected by LC-MS.

Fig. S14. LC-MS evidence for **1e-B-TEMPO**.

Fig. S15. LC-MS evidence for **1e-C-TEMPO**.

3) The trapping experiment for the synthesis of **2a**

Scheme S5. TEMPO trapping experiment for **2a**.

In a 4 mL vial equipped with (E)-1-benzylidene-2-phenylhydrazine (0.1 mmol), 2-bromoacetophenone (0.15 mmol), TEMPO (0.3 mmol), TEA (5 μ L), CsPbBr₃ Perovskite NCs **P1** (2.0 mg), and 1 mL DCM were added and then stirred under the irradiation with blue LED lamp for 24 h. Afford **2a** with 20% yield, along the trapping product was isolated and confirmed by ¹H NMR.

4) The trapping experiment for the synthesis of **2f**

Scheme S6. TEMPO trapping experiment for **2f**.

In a 4 mL vial equipped with ethyl (E)-3-phenyl-3-(phenylamino)acrylate (0.1

mmol), 2-bromoacetophenone (0.15 mmol), TEMPO (0.3 mmol), TEA (5 μ L), CsPbBr₃ Perovskite NCs **P1** (2.0 mg), and 1 mL DCM were added and then stirred under the irradiation with blue LED lamp for 24 h. Two trapping products **2f-A-TEMPO** and **2f-B-TEMPO** were detected by LC-MS.

Fig. S16. LC-MS evidence for **2f-B**.

Fig. S17. LC-MS evidence for the **2f-B-TEMPO**.

2. In the second part of this paper, the author try to reveal the size effect of perovskite on the reaction. However, only two sizes, 6nm and 9nm, were tested. From the

principle of statistics, it is difficult to obtain any information from this limited comparison.

Reply: Thanks for the comment. It is true that two sizes are not enough and we have synthesized two more new batches of QDs materials (4 nm, $\lambda_{\text{PL}} = 467$ nm and 14 nm, $\lambda_{\text{PL}} = 525$ nm) according to the reference (*J. Phys. Chem. Lett.* 2018, 9, 3093-3097) to further prove our conclusion that “small size NCs in general promote a faster reaction rate, but not necessarily a higher yield unless presenting in a perovskite friendly reaction environment”. The UV-vis and PL spectra with these new batches are shown and updated in Fig. 2d, and the comparison for the reactions of **1b** and **2a** are shown in Fig. S4 in supporting information. And the related discussions are updated in the revised manuscript as shown below:

In contrast, using a high temperature synthetic method, we also synthesized size-controlled CsPbBr₃ NCs (**P2**, 4 nm, $\lambda_{\text{PL}} = 467$ nm; **P3**, 6 nm, $\lambda_{\text{PL}} = 508$ nm; **P4**, 9 nm, $\lambda_{\text{PL}} = 515$ nm; **P5**, 14 nm, $\lambda_{\text{PL}} = 521$ nm, Fig. 2b-d and Fig. S3). As shown in Fig. 2d, these NCs show a blue shift probably due to quantum confinements. Their photocatalytic ability has also been explored in the same reaction condition (catalyst loading 1.0 mg). For example, in C-H activation, at the early stage of the reaction, we find that smaller size NCs, i.e. P3-P5 show much higher initial reaction rate compare to the original synthesized P1 NCs. (Fig. S4-5). However, small size NCs' catalytic reactivity diminished quickly. When breaking a C-Br bond to form **1b**, the reaction yield is recorded as 54-64% using P3-P5 in less than 40 min, and longer reaction time leads to a marginal increase of the yield of **1b**. Much lower yield, ~8% was observed within P2 not only because of a faster deactivation using a smaller size, but also a significant blue shift leading to less visible absorption. Whereas using P1, the reaction rate is slower, however, the yield was observed to continuously increase and reach to 85% in ca. 5 hours.

Figure 2. (d) UV-vis and PL spectra of CsPbBr₃ **P2-P5**;

3. The perovskite is instability in CH_2Cl_2 , as the author said it can be attributed to the ion exchange, XRD or XPS is a better method to prove it. “whereas no obvious PL changes are observed in non-halide solvents (Fig. S6).” but I do not see the relevant PL spectrum in Figure S6. In Fig 2f and 2g, the author try to use XRD to prove the ion change during the photocatalytic reaction, but the logic is too complex. It would be more intuitive if the author make a comparison on CsPbBr_3 between before and after reaction.

Reply: We are very sorry for this confusion and thank you for pointing this confused points out. a) As we mentioned in the manuscript, the perovskite is quite stable in CH_2Cl_2 without light; however, with the irradiation of Blue LED, Br/Cl halide exchange happens. This observation is also corroborated with previous work (*J. Am. Chem. Soc.* 2017, 139, 4358-4361; *Catal. Sci. Technol.*, 2018, 8, 4257-4263). Regarding the statement “whereas no obvious PL changes are observed in non-halide solvents (Fig. S6)”, we are very sorry, we only give the **contrast images** under UV lamp in original Figure S6a to indicate there is no change of the PL in hexane under the LED illumination. However, we did not show the PL spectra before and after the irradiation of blue LED. In the revised manuscript we have added the relevant PL spectra in hexane and 1,4-dioxane in Fig. S7c-d. The updated version included the missing information for the non-halogenated solvent as shown in Figure S7.

Fig. S7. (a) The photograph for CsPbBr_3 P3 before and after the irradiation of Blue LED for 2h in Hexane, CH_2Cl_2 and CH_2Br_2 at ambient light (top) and 365 nm UV light (bottom); (b) The UV-vis and PL spectra for P3 before the irradiation of LED

and after irradiation in CH_2Cl_2 for 1h; (c) The PL spectra for **P3** before the irradiation of LED and after irradiation in Hexane for 5h; (d) The PL spectra for **P3** before the irradiation of LED and after irradiation in 1,4-dioxane for 5h.

In addition, regarding the XRD, we are sorry for the confusion and complexity. We demonstrate the comparison before and after reaction in Fig. 2f and 2g. We are sorry for the complexity here. As shown in the updated Figure 2f and 2g, we record the XRD in three different time periods, the top one is the synthesized CsPbBr_3 catalysts before mixing with anything; the middle XRD is the isolated solid after mixing with substrate and co-catalyst but before illumination; the bottom one is the solid isolated after reaction. The middle XRDs indicate a fast anion-change with co-catalyst $(\text{ClCH}_2\text{CH}_2)_2\text{NH}_2\text{Cl}$, forming a $\text{CsPbBr}_x\text{Cl}_{3-x}$ within both reaction 1b and 1c. However, the bottom XRDs are very different, 1b ending with CsPbBr_3 due to the chemical-equivalent byproduct Br^- anions, and 1c ending with CsPbCl_3 due to the chemical-equivalent byproduct Cl^- anions. Related discussions have been added in the manuscript to further clear the confusion.

Fig. 2 (f) XRD of as-prepared CsPbBr_3 **P1**; isolated from the reaction **1b** before and after irradiation; (g) the corresponding XRD for reaction **1c**.

4. The author use acid to enhance the PL intensity and obtain a conclusion that “non-halide organic acid may not only stabilize the perovskite NCs, but also may increase the overall catalytic efficiency for respective reactions (1a, 1d and 1e)”. Can the author use photocatalytic reactions (1a, 1d and 1e) to prove this conclusion?

Reply: Thanks for this comment and we are sorry to miss the discussion of the correlation between amount of non-halide acid and the final catalytic efficiency. In fact, in our Table S1, S4 and S5. We demonstrate that no product or trace amount of the product were afforded without adding TFA. This indicated, in the catalytic mechanism that, the protons are the key component for the catalytic cycle to accomplish. Without acid, many reactions cannot proceed. This is also corroborated with our discussion at the beginning of this paragraph: “Acidity or free protons in perovskite reaction mixture may play a role in organic synthesis. For instance, carboxylic acids such as propionic acid, benzoic acid or trifluoroacetic acid (TFA),

were used either as the co-catalyst (1a, 1d and 1e) or as a substrate (3a-3f)..."

In addition, we also specifically explore the TFA amount with regards to our final catalytic efficiency and yield. Different amount of TFA in photocatalytic reaction (1a, 1d, and 1e) has been explored to prove our conclusion. Such data has been added and updated in the Table S1, S4 and S5. We find that suitable amount of TFA (catalytic amount for **1a**, and ~1.0 equiv. for **1d** and **1e**) is essential for these reactions to proceed. Zero amount or too much amount of TFA in the reaction mixture will result in a significantly decreased reaction yield. This observation is also consistent with the decreasing of PL Intensity in too high concentration of TFA (Fig. 2h).

In this regard, we really appreciate this reviewer's comment to help to improve the quality of such discussion and the manuscript has also been revised below accordingly regarding this comment. "The maximum PL enhancement was observed using TFA at a concentration of *ca.* 6.5-13 mM, more acid leads to a diminishing PL probably because large number of protons may start to initiate a deactivation process. Interestingly, such optimized TFA concentration also leads to a maximum product yield of 1a, 1d and 1e as shown in Table S1, S4 and S5 respectively, indicating a high PL of the photocatalyst may increase the catalytic conversion. Therefore, non-halide organic acid may not only stabilize the perovskite NCs, but also may increase the overall catalytic efficiency for respective reactions."

5. From the stability test in SI, it seems like the stability of perovskite during photo reaction is too poor. P2 seems like loss the activity suddenly. what is happening at that point?

Reply: Thanks for the comment. It is true that the small perovskites particles in the quantum dots level that decompose very quickly and are too poor in reactivity. That is one of the differences that our photocatalytic (**P1**) system comparing to some other reported perovskite results. We noticed that **P3** (previous version noted as **P2**) seems like a sudden loss of reactivity according to Figure S4a. One of the assumptions is that the time intervals are too wide in the current exploration, but the deactivation is too quick. We extracted a data point in every 20 minutes use NMR studies. And we basically find that it lose activity after 2 data points. We would like to know more information, particularly in between 20 min and 40 min. Therefore, we have re-test the ¹H NMR yield with regards to the time using **P3**, and we can extract one more data point in between at 10 min and 30 min. As a result, we find the yield is ~20% at 10 min, and ~50 % at 30 min which is close to saturated line. We re-plot the data points in Figure S4a and no sudden loss like previous figure was observed. Therefore, we do not think there is a sudden catalytic activity lose here, but because our data points are too few, and not enough to demonstrate such change. Hence, we have updated the Figure S4 with more experimental data points, 10, 20, 30, 40 min and now it has been showing that a graduate deactivation process and not a sudden loss, as shown below. We are sorry for this mis-leading information in the first version. And

we indeed thank this reviewer very much for this helpful comment.

Fig. S4. (a) Time dependence for the perovskite **P1-P5** of reaction yields for **1b**; (b) Time dependence for the perovskite **P1, P3** and **P5** of reaction yields for **2a**.

6. I am confused by the author's statement about the ligands. CdSe QD is stabilized by oleic acid, the author's sample still need ligand to stabilize, n-octylammonium, I do not see any advantage about this point.

Reply: we are really sorry for the confusion. This reviewer is right, both CdSe QDs and perovskite need to stabilize by capping ligands during the synthesis. After isolation, the capping ligands are still there. However, when using as a photocatalyst for photoredox reaction, as shown in the reference (*J. Am. Chem. Soc.* 2017, 139, 4250-4253), CdSe QDs need to be further stabilized with adding extra oleic acid and trioctylphosphine as ligand in ODE solution. The catalyst has to be loaded in ODE or in hexanes solution, otherwise reaction is not working. Such stabilization strategy for CdSe is very important and has been discussed exclusively in the paper with various stabilization ligands including A, stearate and trioctylphosphine; B, oleate and trioctylphosphine; C, oleate and diphenylphosphine. And they find the B give the best yield. In order to make this comparison more reasonable, we also conduct experiments. Simple stabilized with oleic acid using commercially available CdSe in 3.0 nm is not working or resulting in trace amount of product (Table S2, S6-S11). However, in this reference, the reason on the ligand's role, particularly phosphine's role has not been discussed/illustrated in this paper.

In our perovskite system, the ligand role is more straightforward, it is used to for the colloidal solution to form in the DCM or other organic solvents. And such capping ligand (ammonium as A site) is part of the $APbX_3$ structure, hence more stable and therefore, we can directly load the catalyst in solid form without any pre-suspension. Here, we added experiments in Table S3-S5 to evaluate the capping ligand's role for our photocatalytic reaction and we find that capping ligand is not altering any catalytic results at all in our system. we revised our manuscript as shown below:

For example, addition of trioctylphosphine and extra oleic acid was found to be

essential to stabilize CdSe QDs during the reported catalytic reactions, otherwise very low or no yield of products can be obtained.³⁹ Discrepancy on various phosphine stabilizing ligands or carboxylic capping ligands leads to significant change on the product yield (12%-70%) using CdSe QDs.³⁹ In addition, introducing extra oleic acid may result in complexity of organic reactions, for example, competition in esterification is observed for reaction **3a**. While changing capping ligand on perovskite plays little role in the final product yield as shown in Table S3-S5. This is probably because the capping ligands (*e.g.*, *n*-octylammonium) that stabilize perovskite colloids are reported to function as A site to the perovskite APbX₃ structure,³¹ hence no extra stabilization protocol is required using perovskite nanocrystal for photocatalysis.

7. In the last part, replacing Br with Cl and I to tune the bandgap has been described in many publications. Some contents are similar to the author's JACS paper.

Reply: Thanks for this comment. Indeed, the bandgap tuning via halides exchanging has been described in many publications, including our recent published JACS paper (briefly mentioned at the end regarding the theoretical possibility or envision regarding the band-tuning's effect on catalytic reaction). However, so far as we know, band tuning's **experimental effects** on photocatalytic reaction have not been discussed, nor explored. Thanks to this reviewer's comments. In this revised version, we have accordingly removed the similar contents on the theoretical envision, but focused on the new **experimental discovery** in this work. To our knowledge, here it is the first time to illustrate that band-tuning of perovskite that can result in significant difference in photocatalytic reaction outcomes. We demonstrate that intentional or in situ band-tuning experiments of CsPbBr₃ NCs successfully drive previously unachievable reactions, *i.e.* reactions showing in Figure 3d, **2i** and **3f** via halide exchange as discussed in our manuscript. Specifically, we first hypothesize that band-tuning via in-situ ion-exchange (PL blue-shift) of perovskite is similar to increase the triplet energy E_T in Ir molecules, thus leading to a higher yield of product. And then we have conducted a systematic band-tuning experiment (via adding known amount of TMSCl to tune the bandgap) to confirm such correlation between the bandedges and the product yield.

More importantly, our hypothesis is transformative and also proved that such band-tuning is very powerful. It may also let the people to **re-think** the general assumption in organic chemistry that C-Cl bond are stronger than C-Br bond and hence harder to activate. For instance, such banding-tuning with halide exchange can activate C-Cl bond towards C-C bond formation reactions. Without band-tuning, a weaker C-Br bond, (α -bromoketone) is actually non-reactive in the same system. We therefore proved that band-tuning may result in an absolute discrepancy in photo-activation reactions.

Overall, we really appreciate the valuable suggestions and comments from the reviewers who helped us to improve the quality of this work. And we are also grateful

for the opportunity to re-submit our manuscript. We hope this revised manuscript is acceptable for publication in “*Nature Communications*”.

Reviewers' comments:

Reviewer #1 (Remarks to the Author):

The points raised in the previous round of review have been satisfactorily addressed by the authors. I would therefore recommend its publication in Nature Communications at its current format.

Reviewer #3 (Remarks to the Author):

I will start by saying that I agree with the premise that perovskites (bulk and nanostructured) should have a large role in photocatalysis for organic synthesis, given their relatively high stability under water-free conditions.

However, this manuscript, while ambitious, reads like a condensed review, not a communication. The problem starts early on, where the questions posed are so broad that one can't possibly be expected to evaluate their answers in a single manuscript. Manuscripts, especially communications, should contain one main thesis -- this manuscript contains 10 or more. I appreciate the scope of work here, but the condensation of so many reactions PLUS characterization of the perovskite nanocrystals under certain conditions (which should clearly be a separate paper) dilutes the discussion so dramatically that I cannot discern the major claims of the paper, other than that perovskites can perform some chemical reactions in high yield. What is missing here is (i) comments about reproducibility, (ii) thorough comparison of critical catalytic parameters (TON, TOF, driving force, etc) with the state-of-the-art, (iii) discussion of catalyst loading, (iv) action spectra proving photocatalytic activity of the material and identifying the contributing populations within clearly heterogeneous samples, (iv) ANY precise discussion of mechanism.

Because of the condensed nature of the discussion, many conclusions are speculative and comparisons to, say QD or molecular photocatalysts, include statements that just aren't true in general. There's also an abundance of imprecise language -- talking about, for example, "more catalytic sites" and "larger surface areas" of smaller NCs rather than the more correct surface area-to-volume ratio. Also the discussion of "deactivation" of the nanocrystals really contains no physical insight.

In summary, the authors make an definite impression that perovskites are promising materials for this application, but the data provided and the discussion do not, in my opinion, constitute a complete scientific study of any one reaction or any one catalytic material, so I cannot recommend this paper for publication in its current form.

Response to Reviewer 1

Reviewer #1

Comments:

The points raised in the previous round of review have been satisfactorily addressed by the authors. I would therefore recommend its publication in Nature Communications at its current format.

Reply: We indeed thank this reviewer very much for the valuable comments and suggestions again.

Response to Reviewer 3

Reviewer #3:

1. I will start by saying that I agree with the premise that perovskites (bulk and nanostructured) should have a large role in photocatalysis for organic synthesis, given their relatively high stability under water-free conditions.

Reply: We indeed thank this reviewer very much for the valuable comments and suggestions and we also would like to revise according to these suggestions and comments.

2. However, this manuscript, while ambitious, reads like a condensed review, not a communication. The problem starts early on, where the questions posed are so broad that one can't possibly be expected to evaluate their answers in a single manuscript. Manuscripts, especially communications, should contain one main thesis -- this manuscript contains 10 or more. I appreciate the scope of work here, but the condensation of so many reactions PLUS characterization of the perovskite nanocrystals under certain conditions (which should clearly be a separate paper) dilutes the discussion so dramatically that I cannot discern the major claims of the paper, other than that perovskites can perform some chemical reactions in high yield.

Reply: Thank you for your comments on this point. We agree that too many reactions indeed distract the main thesis of this manuscript. And many certain condition discussions can be a separate paper. Accordingly, we focused on three types general reactions, C-C, C-N and C-O and also remove the distracted reactions: halide reduction, polymerization reactions. We note that these two types of reactions have not been actually discussed too much in the previous version at all. Inspiring by this reviewer's comment, we also note that the reaction **1a** and **3g-3i** in the old version also have NOT been thoroughly discussed. Although adding these reactions into the discussion seems likely expanding the perovskite's general catalytic ability or scope of the work towards photocatalysis, yet they do not provide essential points toward understanding the key chemistry here but introduce distractions. We take the advice and remove these distracted reactions, but still keep the three fundamental types of reactions in the

category of C-C, C-N and C-O bond formation reactions. In this way, we may not only demonstrate the general acceptance of our perovskite towards photocatalytic activation of three fundamental types of organic reactions, but also present in a neat way to highlight the key catalytic parameters, mechanisms and unique properties of perovskite. In following discussion after Figure 1, we also categorized such reactions into three types and focus on discussion of mechanisms using perovskite comparing to other photocatalysts. We hope our revised manuscript is in an acceptable manner to demonstrate the key point that the Lead Halide Perovskites is truly a potential photocatalyst candidate for general acceptance in fundamental organic synthesis.

3. What is missing here is (i) comments about reproducibility, (ii) thorough comparison of critical catalytic parameters (TON, TOF, driving force, *etc*) with the state-of-the-art, (iii) discussion of catalyst loading, (iv) action spectra proving photocatalytic activity of the material and identifying the contributing populations within clearly heterogeneous samples, (v) ANY precise discussion of mechanism.

Reply: We thank this reviewer very much for these insightful comments on the key parameters. These comments indeed helped us to improve the quality of this work. Accordingly, here in the revised manuscript we have added the discussion of the following:

(i) Reproducibility.

To test the reproducibility, we conducted each type of reactions, **1a**, **1c**, **1d**, **2a**, **2g**, **3f** for three times in their corresponding optimized conditions using the solvent and materials treatment in SI. The respective data has been updated in Fig. 1 and Table S8 was also listed in the new version. The results show that our photocatalyst leads to a reasonable reproducibility for C-C, C-N, C-O bond formation reactions under such optimized conditions.

Table S8. Reproducibility of the photocatalytic reactions with **P1** in their corresponding optimized conditions.^a

	Yield (%) ^b					
	1a	1c	1d	2a	2g	3f
1st	85	90	79	86	93	72
2nd	82	81	80	82	79	67
3rd	86	86	71	84	84	71
Average	84	85	76	84	85	70

^a Conditions: Table S1, entry 2 for **1a**; Table S3, entry 1 for **1c**; Table S4, entry 1 for **1d**; Table S5, entry 1 for **2a**; Table S6, entry 1 for **2g**; Table S7, entry 1 for **3f**.

^b Yields determined by ¹H NMR.

1. C-C bond formation via C-H activation

2. *N*-Heterocyclization

3. C-O cross-coupling

Figure 1. The library of C-C, C-N and C-O bond formation reactions and respective yield. (Yields of **1a**, **1c**, **1d**, **2a**, **2g**, **3a** are the average yields of three times reactions details in Table S8; Inset: perspective view of **1d**'s single crystal structure with the thermal ellipsoids drawn at 50% probability level and the H atoms omitted for clarity.)

ii) Critical catalytic parameters of TON and driving force have been explored and their comparison with reported photocatalysts have also been illustrated as followed. Accordingly, we have explored the TON as suggested and the following part has been updated in the manuscript as well as in SI:

“Catalytic turnover number (TON) is compared and listed in Table 1. Heterogeneous catalyst, *i.e.* 3.0 nm CdSe QDs were reported to optimally render a TON of 79,100 (based on QD's molecular weight Mw, 88,000 g/mol) in glove box.³⁹ However, in our condition under air, no yield (nor TON) of **1**, **2** and **3** can be obtained using CdSe QDs. In addition to air-sensitivity, CdSe's performance was also dependent on size and capping ligands.³⁹ While changing capping ligand on perovskite plays little role in the yield as shown in Table S2-S4. This is probably because the capping ligands (*e.g.*, *n*-octylammonium) that stabilize perovskite colloids are reported to function as A site to the perovskite APbX₃ structure,³¹ hence no extra stabilization protocol is required using perovskite nanocrystal for photocatalysis. Using the method in CdSe QDs³⁹ to calculate TON, **P2** NCs (14 nm, based on Mw, 8,015,000 g/mol, **P1-P5** TON details see Table S9) renders 2,565,000. Perovskites' heterogeneous catalytic ability is

validated via regaining strong PL after recovering the catalyst via centrifuge after reaction (Fig. S7). To compare TON with molecular catalysts, TON calculation based on mole of metal (independent of size, CsPbBr₃, 579.8 g/mol) was carried out instead. For instance, four cycles of the reactions render a TON of at least 9,100 for **1a** (Table 1, details see SI). Overall, one or two orders of higher TONs under our condition are observed using perovskite than others, except reaction **3**, in which TON may rely on both perovskite and Ni co-catalyst.”

Table 1. Comparison of photocatalysts for corresponding reactions in air or in oxygen.

Photocatalyst ^a	Yield (%) ^b					TON (based on CsPbBr ₃) ^c				
	1a	1c	1d	2a	3f	1a	1c	1d	2a	3f
CsPbBr ₃ P1	84	85	76	84	70	9100	830	280	380	33
Ru(bpy) ₃ (PF ₆) ₂	Trace	60	25	Trace	N.R.	-	60	25	-	-
Ir(ppy) ₃	79	N.R.	N.R.	63	65	79	-	-	63	33
CdSe QDs (525 nm)	Trace	Trace	N.R.	N.R.	N.R.	-	-	-	-	-

^a bpy = 2,2'-bipyridine; ppy = ortho-metalated 2-phenylpyridine; ^b average yield using for **P1**; ^c details in SI.

TON of **1a**:

For a 5.0 mmol scale (10 times scale) reaction **1a**, 1.0 mg **P1** was used, leading to **1a** in 82% yield after 16 hours. After centrifuging the reaction mixture to recover perovskite and to re-apply for a same scale reaction, catalyst is still robust after at least four-times repeats (yield 82%, 80%, 79%, 73%, respectively). TON of CsPbBr₃ catalysis is calculated in this way: TON is equal to total mol of product **1a** in 4 repeats of reaction over mol of Pb. $TON = (5.0 \text{ mmol} \times (0.82 + 0.80 + 0.79 + 0.73)) / (1.0 \text{ mg} / 579.8 \text{ g/mol}) \approx 9100$.

TON of **1c**:

For a large-scale reaction of **1c** (2.0 mmol), 1.0 mg **P1** was used, leading to **1a** in 76% yield after 24 hours. $TON = (2.0 \text{ mmol} \times 0.71) / (1.0 \text{ mg} / 579.8 \text{ g/mol}) \approx 830$.

TON of **1d**:

For a large-scale reaction of **1d** (2.0 mmol), 1.0 mg **P1** was used, leading to **1a** in 49% yield after 24 hours. $TON = (2.0 \text{ mmol} \times 0.49) / (2.0 \text{ mg} / 579.8 \text{ g/mol}) \approx 280$.

TON of **2a**:

For a 0.2 mmol scale reaction **2a**, 1.0 mg **P1** was used, leading to **2a** in 80% yield after 8 hours. After centrifuging the reaction mixture to recover perovskite and to re-apply for a same scale reaction, catalyst is still robust after at least four-times repeats

(yield 83%, 83%, 81%, 79%, respectively). $\text{TON} = (0.2 \text{ mmol} \times (0.83 + 0.83 + 0.81 + 0.79)) / (1.0 \text{ mg} / 579.8 \text{ g/mol}) \approx 380$.

TON of **3a**:

For a 0.4 mmol scale reaction **3a**, 5.0 mg **P1** was used, leading to **3a** in 76% yield after 48 hours. $\text{TON} = (0.4 \text{ mmol} \times 0.70) / (5.0 \text{ mg} / 579.8 \text{ g/mol}) = 33$.

Driving force:

“To further elucidate the reaction mechanism, electrochemical studies were conducted. (Fig. S24-31) According to the comparison between redox potentials of the key substrates and the band energy of perovskite, the respective driving force is listed in Figure 3. Driving force for HT in reaction **1c**, **1d** and **2a** is observed among ~ 0.1 to 0.3 eV , consistent with the Stern-Volmer quenching results (Fig S11-17) as well as the mechanistically verified intermediates in Scheme 1. However, **2g-B** disfavors HT due to a more positive oxidation potential (E_{ox} , 1.42 V), corroborating with the previous observation that direct radical forming from **2g-B** is difficult, unlike reaction **2a** pathway. Moreover, driving force for ET is listed from ~ 0.2 to 0.5 eV , confirming our discussion on ET in Scheme 1. However, noticeable exception, 2,4'-dichloroacetophenone, though presenting a more negative reduction potential (E_{red} , -1.47 V), reacts to form respective pyrrole. We postulate that *in-situ* band-tuning of perovskite may play a role here and is discussed below.”

Figure 3. Band energy of CsPbBr₃ vs the redox potentials of substrates.

Determination of Redox potentials of key substrates.

Fig. S24. CV spectra of 2-Bromoacetophenone using NBu_4PF_6 as electrolyte in degassed CH_3CN , $[\text{NBu}_4\text{PF}_6] = 0.1$ M. $E_{\text{red}} = -1.11$ V vs SCE.

Fig. S25. CV spectra of 2-bromo-4'-nitroacetophenone using NBu_4PF_6 as electrolyte in degassed CH_3CN , $[\text{NBu}_4\text{PF}_6] = 0.1$ M. $E_{\text{red}} = -0.74$ V vs SCE.

Fig. S26. CV spectra of 3-(bromoacetyl)coumarin using NBu_4PF_6 as electrolyte in degassed CH_3CN , $[\text{NBu}_4\text{PF}_6] = 0.1$ M. $E_{\text{red}} = -0.92$ V vs SCE.

Fig. S27. CV spectra of 2,4'-dichloroacetophenone using NBu_4PF_6 as electrolyte in degassed CH_3CN , $[\text{NBu}_4\text{PF}_6] = 0.1$ M. $E_{\text{red}} = -1.47$ V vs SCE.

Fig. S28. CV spectra of 4-phenylmorpholine using NBu_4PF_6 as electrolyte in degassed CH_3CN , $[\text{NBu}_4\text{PF}_6] = 0.1 \text{ M}$. $E_{ox} = 0.89 \text{ V vs SCE}$.

Fig. S29. CV spectra of 2-phenyl-1,2,3,4-tetrahydroisoquinoline using NBu_4PF_6 as electrolyte in degassed CH_3CN , $[\text{NBu}_4\text{PF}_6] = 0.1 \text{ M}$. $E_{ox} = 0.90 \text{ V vs SCE}$.

Fig. S30. CV spectra of (*E*)-1-benzylidene-2-phenylhydrazine using NBu_4PF_6 as electrolyte in degassed CH_3CN , $[\text{NBu}_4\text{PF}_6] = 0.1 \text{ M}$. $E_{\text{ox}} = 1.00 \text{ V vs SCE}$.

Fig. S31. CV spectra of ethyl (*E*)-3-phenyl-3-(phenylamino)acrylate using NBu_4PF_6 as electrolyte in degassed CH_3CN , $[\text{NBu}_4\text{PF}_6] = 0.1 \text{ M}$. $E_{\text{ox}} = 1.42 \text{ V vs SCE}$.

(iii) Catalyst loading exploration has also been explored and added in the revised manuscript as shown in Figure 1 as well as in the discussion context of the manuscript.

The loading exploration and their comparison has been also showing in Table S1-S7.

“Catalyst loading has also been explored (Table S1-S7) and respective minimum loading for typical reactions of ~ 0.1-0.5 mmol has been listed in Fig. 1. These reactions result in respective products in moderate to high yields *without* need for anaerobic sparging.”

(iv) We have conducted further spectroscopy investigation including PL and their related Stern-Volmer quenching studies to illustrate the heterogeneous nature of NCs catalyst. As shown in **Fig. S7**, we found that the recycled photocatalyst CsPbBr₃ (after a completely reaction cycle) in EtOAc is still emissive. Recycling of these NCs via centrifuge (inset) indicated the strong PL of the perovskite. Reloading such catalyst leads to the second cycle of the catalytic reactions. Moreover, the Stern-Volmer quenching studies also support our proposed mechanism as shown below.

Fig. S7. The comparison of PL spectra of the initial CsPbBr₃ NCs suspension in EtOAc (before mixing with any substrate, black line); CsPbBr₃ NCs suspension mixed with 1-benzylidene-2-phenylhydrazine, 2-bromoacetophenone and base before the irradiation of LED (red line); after the irradiation of LED for 12h (blue line); the recycled CsPbBr₃ after centrifuging the reaction mixture and re-suspension in EtOAc (purple line); the recycled CsPbBr₃ applied for 2nd time reaction (green line); the 2nd time recycled CsPbBr₃ after centrifuging the reaction mixture and re-suspension in EtOAc (violet line). Inset: photograph of the recycled CsPbBr₃ (containing base residue) after EtOAc washing under UV light.

Fig. S11. CsPbBr₃ NCs Emission quenching by 2-phenyl-1,2,3,4-tetrahydroisoquinoline. $k_q = 1.8 \times 10^8 \text{ M}^{-1}\text{s}^{-1}$.

Fig. S12. CsPbBr₃ NCs Emission quenching by 4-phenylmorpholine. $k_q = 3.6 \times 10^8 \text{ M}^{-1}\text{s}^{-1}$.

Fig. S13. CsPbBr₃ NCs Emission quenching by benzylidenemalononitrile. $k_q = 3.4 \times 10^8 \text{ M}^{-1}\text{s}^{-1}$.

Fig. S14. CsPbBr₃ NCs Emission quenching by (E)-1-benzylidene-2-phenylhydrazine. $k_q = 8.8 \times 10^9 \text{ M}^{-1}\text{s}^{-1}$.

Fig. S15. CsPbBr₃ NCs Emission quenching by ethyl (E)-3-phenyl-3-(phenylamino)acrylate. $k_q = 4.9 \times 10^9 \text{ M}^{-1}\text{s}^{-1}$.

Fig. S16. CsPbBr₃ NCs Emission quenching by 2-bromoacetophenone in 1,4-dioxane. $k_q = 8.8 \times 10^8 \text{ M}^{-1}\text{s}^{-1}$.

Fig. S17. CsPbBr₃ NCs PL changing by adding 2,4'-dichloroacetophenone. No quenching was observed, indicating the initial ET transfer is difficult corroborating with the electrochemical driving force studies.

(v) mechanism.

We have spent most of time in this period to explore the key mechanism of these fundamental reactions. In fact, the exploration of the first four points have substantially helped us to address the last key point: mechanism. For example, the driving force is indeed helping us to understand the charge transfer process as the key step for electron-transfer (ET) or hole-transfer (HT). We have thoroughly conducted the electrochemical experiments to illustrate such driving force. First, for the mechanism exploration, we have distinguished oxygen's role via its presence or absence in C-C bond formations.

More importantly, during the revision, we have also distinguished the mechanism in C-N bond formations, via comparison of the initial radical formation pathway. The driving force is corroborating with most of the key steps that we proposed in terms of radical formation in scheme 1. Such study demonstrated the differences of possible HT between the formation of pyrrole and pyrazole due to the different oxidation level of substrates. Such driving force result (a completely different observation on HT for reaction **2a** and **2g**) encouraged us to further explore the key intermediate because there must be something different between pyrrole and pyrazole formations. *Interestingly, but not surprisingly, we do successfully identify another key intermediate, directly HT transfer forming 2a-C (instead of 2g-C) via TEMPO-trapping experiment.* Such result has been added in manuscript and in SI as well shown as below. Here we really appreciate reviewer 3's comments that help illustrate the different pathway of pyrrole and pyrazole formations.

4) The trapping experiment of 2a-C.

Scheme S5. TEMPO trapping experiment for 2a-C-TEMPO.

In a 4 mL vial equipped with (E)-1-benzylidene-2-phenylhydrazine (0.1 mmol), TEMPO (0.2 mmol), CsPbBr₃ Perovskite NCs P1 (2.0 mg), and 1 mL CH₂Cl₂ were added and then stirred under the irradiation with blue LED lamp for 24 h. The trapping products 2a-C-TEMPO was detected by LC-MS.

Fig. S21. LC-MS evidence for 2a-C-TEMPO.

Overall, according to TEMPO trapping experiment on the key intermediates, the Stern-Volmer quenching results, and the determined redox potentials of the key substrates and possible driving force for the charge transfer, we rationalized our mechanism as followed.

Scheme 1. (a) Proposed mechanisms for the synthesis of **1c** and **1d**; (b) **2a** and **2g**. (Blue square: isolated and characterized by ¹H-NMR; Red square: trapped and detected by LC-MS (Fig. S18-23); HT = hole-transfer; ET = electron-transfer)

“Mechanism. Oxygen may be of an essential component in certain photoredox reactions. For instance in Scheme 1a, radical addition product **1c** is achieved in nitrogen atmosphere while in a similar setup, air or oxygen atmosphere produces a ring-closure **1d** (crystal structure provided in Fig. 1). Oxygen is found to be the key reagent as the hydrogen atom acceptor that further induced the C-H activation on phenyl rings.^{40,41} As shown in Scheme 1, the reaction mechanisms are proposed in which the key radical intermediates have been investigated. Upon Stern-Volmer PL quenching studies (Fig. S11-17), perovskite PL quenching by **1d-A** was observed ($k_q = 3.6 \times 10^8 \text{ M}^{-1}\text{s}^{-1}$, Fig. S12) and resulted in **1d-B** radical in the presence of oxygen. Intermediate **1d-B** and **1d-C** have been verified via radical trapping experiment employing 2,2,6,6-tetramethyl-1-piperidinyloxy (TEMPO) as a radical scavenger, through LC-MS (Fig. S19-20). In the absence of oxygen, radical **1c-B** is also confirmed by TEMPO-trapped product (Fig. S18) and further verified by the self-coupling diastereoisomers **1c-C** via ¹H NMR (see SI). It is worth mentioning that the presence of air leads to more **1c-C** formation and ultimately diminishes the yield of **1c**.

Scheme 1b shows the proposed mechanism of C-N formations, in which both oxidative (ET, **2a-A**) and reductive quenching product (HT, **2a-B**) in reaction **2a** have been trapped by TEMPO (either observed via ¹H NMR or LCMS), indicating a strong charge separation and transfer ability induced by perovskite. This pathway is similar to our previous mechanism exploration in α -alkylation of aldehydes.¹⁷ Radical coupling between **2a-A** and **2a-C** leads to the intermediate of **2a-D**. Then C-N formation via intramolecular cyclization and a final dehydration leads to the pyrazole product **2a**. In contrast, the radical formation from **2g-B** via direct HT has not been observed, instead **2g-C** was verified via radical-trapping, likely demonstrating a different mechanism of

pyrrole formation as shown in Scheme 1b. Mechanism of reaction **3a** is also proposed and shown in Scheme S1 similar to previous reported mechanism.⁴⁵

To further elucidate the reaction mechanism, electrochemical studies were conducted. (Fig. S24-31) According to the comparison between redox potentials of the key substrates and the band energy of perovskite, the respective driving force is listed in Figure 3. Driving force for HT in reaction **1c**, **1d** and **2a** is observed among ~ 0.1 to 0.3 eV, consistent with the Stern-Volmer quenching results (Fig S11-17) as well as the mechanistically verified intermediates in Scheme 1. However, **2g-B** disfavors HT due to a more positive oxidation potential (E_{ox} , 1.42V), corroborating with the previous observation that direct radical forming from **2g-B** is difficult, unlike reaction **2a** pathway. Moreover, driving force for ET is also listed from ~0.2 to 0.5 eV, confirming our discussion on ET in Scheme 1. However, noticeable exception, 2,4'-dichloroacetophenone, though presenting a more negative reduction potential (E_{red} , -1.47V), still reacts to form respective pyrrole. We postulate that *in-situ* band-tuning of perovskite may play a role here and is discussed below.”

4. Because of the condensed nature of the discussion, many conclusions are speculative and comparisons to, say QD or molecular photocatalysts, include statements that just aren't true in general. There's also an abundance of imprecise language - talking about, for example, “more catalytic sites” and “larger surface areas” of smaller NCs rather than the more correct surface area-to-volume ratio. Also the discussion of “deactivation” of the nanocrystals really contains no physical insight.

Reply: We really appreciate this reviewer's comments in this regards that would certainly help us to improve the quality of this work.

Imprecise language regarding “the catalytic site” and “surface area” has also been corrected with surface area-to-volume ratio. And the discussion of “deactivation” has been revised accordingly. Such details are listed in SI and as showed below.

Calculation of NC cubic edge length d for P2-P5:

According to the well-established size-dependent absorbance spectrum of CsPbBr₃ NC from ref S3.

$$E_g(d) = E_g(\infty) + \frac{1}{(a+bd+rd^2)}$$

Where $E_g(d)$ comes from first exciton of absorption, $E_g(\infty) = 2.25$ eV, $a = -1.26$ eV⁻¹, $b = 0.996$ nm⁻¹eV⁻¹, $r = -0.0324$ nm⁻²eV⁻¹.

Calculation of numbers of CsPbBr₃ units in a NC.

$$\text{NC size/lattice of CsPbBr}_3 = (d/0.583)^3$$

Calculation of Molecular weight of a NC.

$$\text{Molecular weight} = (d/0.583)^3 \times 579.8 \text{ (g/mol)}$$

Calculation of TON based on Molecular weight of NC

$$\text{Moles of desired product formed/moles of catalyst} \\ = n_{\text{product}} / \{m_{\text{NC}} / [(d/0.583)^3 \times 579.8]\}$$

Calculation of Surface area-to-volume ratio:

$$\text{Surface area of one NC} = 6d^2; \text{ volume of a NC} = d^3 \text{ (nm}^3\text{)}$$

$$\text{Thus, surface area-to-volume ratio: } 6/d \text{ (nm}^{-1}\text{)}$$

Table S9. Key parameters for CsPbBr₃ **P1-P5**.^a

NCs	First exciton peak (nm)	$E_g(d)$ (eV)	NC size d (nm)	Molecular weight of NC (g/mol) ^b	Surface area-volume ratio (nm ⁻¹)	TON for 1a ^c (based on molecular weight)
P1			24 ^b	39,960,000	0.25	16,983,000
P2	515	2.41	14.0	8,015,000	0.43	2,565,000
P3	506	2.45	8.8	2,375,000	0.68	677,000
P4	490	2.53	6.0	580,000	1.00	157,000
P5	456	2.72	3.9	199,000	1.54	7900

^a TON calculation method using ref 39.

^b Size of **P1** was an estimated average size from TEM image.

^c Yields from **Fig. S3a**.

We have also carefully revised speculative comment and comparisons. And in the new version, particularly regarding discussion of QDs, we have revised as below:

“Catalytic turnover number (TON) is compared and listed in Table 1. Heterogeneous catalyst, *i.e.* 3.0 nm CdSe QDs were reported to optimally render a TON of 79,100 (based on QD’s molecular weight Mw, 88,000 g/mol) in glove box.³⁹ However, in our condition under air, no yield (nor TON) of **1**, **2** and **3** can be obtained using CdSe QDs. In addition to air-sensitivity, CdSe’s performance was also dependent on size and capping ligands.³⁹ While changing capping ligand on perovskite plays little role in the yield as shown in Table S2-S4. This is probably because the capping ligands (*e.g.*, *n*-octylammonium) that stabilize perovskite colloids are reported to function as A site to the perovskite APbX₃ structure,³¹ hence no extra stabilization protocol is required using perovskite nanocrystal for photocatalysis. Using the method in CdSe QDs³⁹ to calculate TON, **P2** NCs (14 nm, based on Mw, 8,015,000 g/mol, **P1-P5** TON details see Table S9) renders 2,565,000. Perovskites’ heterogeneous catalytic ability is validated via regaining strong PL after recovering the catalyst via centrifuge after reaction (Fig. S7). ...”

5. In summary, the authors make a definite impression that perovskites are promising materials for this application, but the data provided and the discussion do not, in my

opinion, constitute a complete scientific study of any one reaction or any one catalytic material, so I cannot recommend this paper for publication in its current form.

Reply: We really appreciate this reviewer's valuable comments that indeed helped us to understand the mechanism and help us to improve the quality of manuscript. We hope that the revised work is at a stage to support publication in *Nature Communications*.

REVIEWERS' COMMENTS:

Reviewer #3 (Remarks to the Author):

This manuscript is still a bit unwieldy, but an improvement over the previous round. I think it's publishable although I recommend the authors be more precise about comparison to other systems like CdSe QDs. Charge extraction from CdSe QD is also extremely fast. Yes, those QDs trap carriers, but this depends on surface coverage and identity of ligands, and is not intrinsic to the CdSe material. The authors should consider the band-edges of the perovskites relative to their ligands when discussing the probabilities of charge trapping.

Reviewer #3 (Remarks to the Author):

This manuscript is still a bit unwieldy, but an improvement over the previous round. I think it's publishable although I recommend the authors be more precise about comparison to other systems like CdSe QDs. Charge extraction from CdSe QD is also extremely fast. Yes, those QDs trap carriers, but this depends on surface coverage and identity of ligands, and is not intrinsic to the CdSe material. The authors should consider the band-edges of the perovskites relative to their ligands when discussing the probabilities of charge trapping.

Reply: We're very glad to see this reviewer's support on the publication of our work. And we thank this reviewer very much for the comment. We accept this reviewer's recommendation that we should be more precise in the charge extraction comparison to other systems like CdSe QDs. Charge extraction from CdSe QD according to reference (ref 47) is indeed also very fast. So this reviewer is definitely right here. We also really appreciated this reviewer's comment that "Yes, those QDs trap carriers, but this depends on surface coverage and identity of ligands, and is not intrinsic to the CdSe material." The detailed reason why perovskite system here behaves better than CdSe under current condition, may contain a lot of reasons. In addition to the comparison of air/oxygen quenching in our manuscript, ligand coverage and identity of ligands in ref 39, there might be also hole trapping effect (due to the ligands type and surface coverage, as this reviewer mentioned) that may cause photocatalysis difference in performance. At this point, we feel further discussion/comparison of the reason to cause such charge trapping between CdSe and perovskite, particularly hole trapping, is beyond the scope of this work. Here, we would like to focus on illustrating that lack of electron and hole traps and fast interfacial electron transfer and hole transfer rates in perovskite, are imperative that may enable highly efficient perovskite induced photocatalysis. However, we do agree that it is definitely worth investigating on the reasons to cause hole trapping, particularly their difference between CdSe (ligand, surface coverage) and perovskite, and in fact, this is one of the focuses of our future work. Therefore, we take this reviewer's advice and remove the imprecise comparison of CdSe with perovskite on charge extractions and trapping. In the revised manuscript, we remove such comparison between CdSe and perovskite in terms of charge extraction and trapping.